# Modeling Interference for Treatment Effect Estimation in Network Dynamic Environment

**Qiang Huang,**
Department of Machine Learning
Mohamed bin Zayed University of
Artificial Intelligence
Abu Dhabi, UAE
qiang.huang@mbzuai.ac.ae

**Jin Tian,**
Department of Machine Learning
Mohamed bin Zayed University of
Artificial Intelligence
Abu Dhabi, UAE
jin.tian@mbzuai.ac.ae

## Abstract

In recent years, estimating causal effects of treatment on the outcome variable in network environments has attracted growing interest. The intrinsic interconnectedness of network and the attendant violation of the SUTVA assumption have prompted a wave of treatment effect estimation methods tailored to network settings, yielding considerable progress such as capturing hidden confounders by leveraging auxiliary network structure. Nevertheless, despite these advances, the existing methods: (*i*) mainly focus on the static network, overlooking the dynamic nature of many real-world networks and confounders that evolve over time; (*ii*) assume the absence of dynamic network interference where one unit's treatment can affect its neighbors' outcomes. To address these two limitations, we first define a new estimand of treatment effects accounting for interference in a dynamic network environment, i.e., CATE-ID, and establish its identifiability under such an environment. Then we accordingly propose DSPNET , a framework tailored specifically for treatment effect estimation in dynamic network environment, that leverages historical information and network structure to capture time-varying confounders and model dynamic interference. Extensive experiments demonstrate the superiority of our proposed method compared to state-of-the-art approaches.

## 1 Introduction

Treatment effect estimation (Winship & Morgan, 1999) plays a fundamental role in understanding the relationship between treatment (a.k.a intervention) and outcome, serving as a cornerstone for decision-making across a wide range of domains, such as healthcare (Hernán & Robins, 2006), social networks (Barabas & Jerit, 2009), and economics (Lechner et al., 2011). The gold standard for estimating treatment effects is randomized controlled trials (RCTs) (Stanley, 2007; Stolberg et al., 2004), which involve randomly assigning units to different treatment arms (e.g., administering or withholding medication) and comparing their observed outcomes (e.g., recovery rates). This design enables reliable estimation of treatment effects by eliminating confounding through randomization. However, RCTs are often costly and time-consuming (Bondemark & Ruf, 2015; Zabor et al., 2020), and may raise ethical concerns in certain contexts (Edwards et al., 1999).

Given the limitations of RCTs, researchers have increasingly turned to rich and readily available observational data to estimate the conditional average treatment effect (CATE)[1]. Many observational studies (Gianicolo et al., 2020; Hernán & Robins, 2006; Rubin, 2007; Winship & Morgan, 1999) rely on the Stable Unit Treatment Value Assumption (SUTVA) (Green & Gerber, 2010), which assumes that units are independent and not subject to interference from one another. Furthermore, one of the critical problems in estimating treatment effects is to eliminate confounding bias caused by confounders (Greenland et al., 1999), the variables that causally affect both treatment and outcome. To facilitate unbiased estimation, these studies also commonly adopt the assumption of ignorability (Greenland & Mansournia, 2015), which assumes the absence of unobserved confounders.

---

[1]While much of the literature equates CATE with Individual Treatment Effect (ITE), in this work we strictly use CATE, as ITE is not identical to CATE theoretically (Vegetabile, 2021).

However, the assumptions of no unmeasured confounding and no interference often fail in practice (Guo et al., 2020a; Yao et al., 2021). In a community-level infectious disease study, socioeconomic status (SES) may affect both treatment (e.g., compliance with mobility restriction) and outcome (e.g., infection risk). For instance, lower-SES individuals may have limited access to safe transportation and are more likely to live in high-density housing, where close contact increases infection risk. However, SES, acting as a confounder, is often difficult to observe or measure directly. To address this, recent work (Chu et al., 2021; Jiang & Sun, 2022; Sui et al.; Veitch et al., 2019) leverages network structures to infer hidden confounders from relational context, such as inferring SES from the occupations or attributes of social connections. Moreover, individuals are embedded in networks where one unit's compliance with mobility restriction can affect the infection risk of its neighbors, a phenomenon known as *interference* or the *spillover effect* (Benjamin-Chung et al., 2018; Ma & Tresp, 2021). Several studies (Huang et al., 2023; Rakesh et al., 2018; Zhao et al., 2024) have been made to explicitly model interference for treatment effect estimation.

However, most of these methods are designed for static networks, assuming that the network structure and covariates remain unchanged over time. In practice, many network environments are inherently dynamic, with both the network structure and the attributes of individual nodes evolving over time. For example, in the aforementioned infectious disease study, the community structure may shift as residents relocate, while individual-level covariates such as health status may also vary over time. These dynamic characteristics pose substantial challenges for treatment effect estimation in networked settings. First, the interplay between complex temporal evolution and network interference makes the identifiability of treatment effects, i.e., determining whether treatment effects can be recovered from observational data, a highly non-trivial problem. Second, as both network structures and node covariates evolve, the distribution of confounders becomes time-dependent; modeling the evolution of confounders and controlling for time-varying confounding bias require further exploration. Third, the evolution of network and attributes alters the pattern and magnitude of interference between nodes, necessitating dynamic modeling of spillover effects based on the changing network structure.

To address the aforementioned challenges, we first define a new target estimand, Conditional Average Treatment Effects with Interference under Dynamic networks (CATE-ID), for treatment effect estimation in dynamic network environments with interference, and formally prove its identifiability under a set of assumptions. Building on this theoretical foundation, we propose DSPNET, a novel framework designed to estimate the target causal estimand by explicitly modeling both hidden confounders and interference in dynamic networks. Specifically, DSPNET integrates GCNs and RNNs to aggregate neighborhood and historical information to infer dynamic hidden confounders. Then it learns a dedicated interference representation to capture spillover effects by encoding the treatments and characteristics of a unit's neighborhood. Finally, DSPNET employs an adversarial learning strategy, encouraging balanced confounder representations to mitigate confounding bias when estimating treatment effects from an observational dynamic network.

## 2 PROBLEM FORMULATION

In this work, we use bold letters to denote vectors or matrices, and unbold lowercase letters for scalars. Specifically, let unbold capital letters denote random variables (e.g., $X_i^t$), lowercase letters (e.g., $\boldsymbol{x}_i^t$) for their realizations. Let $\boldsymbol{A}^t \in \{0,1\}^{N \times N}$ denote the adjacency matrix that encodes the network structure among $N$ units at time step $t$, where $\boldsymbol{A}_{ij}^t = 1$ ($\boldsymbol{A}_{ij}^t = 0$) indicates the presence (absence) of an edge between unit $i$ and $j$. $\boldsymbol{X}^t = \{\boldsymbol{x}_1^t, ..., \boldsymbol{x}_N^t\}$ represents the covariates of the $N$ units at time step $t$ with $\boldsymbol{x}_i^t \in \mathbb{R}^m$ denoting the covariates for unit $i$, and $\boldsymbol{X}^{<t}$ denotes the covariates before time $t$ for all nodes. Let $\boldsymbol{D^t} = \{d_1^t, ..., d_N^t\}$ denote the set of treatment assignments for the $N$ units where $d_i^t = 1$ indicates that unit $i$ is treated and $d_i^t = 0$ otherwise, and $\boldsymbol{Y}^t = \{y_1^t, ..., y_N^t\}$ denotes the set of observed outcomes for $N$ units at time step $t$. Let $\mathcal{G}_i^t$ represent the neighboring set of $i$ at time step $t$ and $\mathcal{G}_{-i}^t$ represent unit $i$'s non-neighbors, and the covariates and treatments of unit $i$'s neighbors and non-neighbors at time step $t$ are denoted by $\boldsymbol{X}_{\mathcal{G}_i}^t, \boldsymbol{D}_{\mathcal{G}_i}^t$ and $\boldsymbol{X}_{\mathcal{G}_{-i}}^t, \boldsymbol{D}_{\mathcal{G}_{-i}}^t$, respectively.

For the treatment effect estimation in I.I.D setting, the target estimand is typically the conditional average treatment effect (CATE): $\tau(\boldsymbol{x}_i) = \mathbb{E}[Y_i(1)|\boldsymbol{x}_i] - \mathbb{E}[Y_i(0)|\boldsymbol{x}_i]$ where $Y_i(D_i)$ denote unit $i$'s potential outcome under treatment $D_i$. However, the above estimand is not applicable in a network environment where interference exists. To account for interference, prior works (Forastiere et al., 2021; Ma & Tresp, 2021) often aggregate the treatments of neighboring units into a single scalar variable (e.g., via mean pooling) and treat it as a regular covariate. However, such a one-dimensional

summary can be inadequate in high-dimensional environments, where it may fail to capture the rich heterogeneity of neighbors' influences. Therefore, we define the following generalized factor that captures the influence of neighbors' covariates and treatment assignments:

**Definition 2.1.** *Environment Exposure.* We define a summary function that aggregates the treatments and covariates of unit $i$'s neighbors at time step $t$, $F_i^t(\cdot): \{\mathbb{R}^m\}^{|\mathcal{G}_i^t|} \times \{0,1\}^{|\mathcal{G}_i^t|} \to \mathcal{E}$ where $\mathcal{E} \in \mathbb{R}^k$ is the exposure space, then the environment exposure is formulated by $E_i^t = F_i^t\big(\boldsymbol{X}_{\mathcal{G}_i}^t, \boldsymbol{D}_{\mathcal{G}_i}^t\big)$.

To ensure well-defined potential outcomes under exposure $E_i^t$, we adopt the following assumption:

**Assumption 2.2.** Given the summary function $F_i^t$, $\forall \, \boldsymbol{D}_{\mathcal{G}_i}^t, \boldsymbol{D}_{\mathcal{G}_{-i}}^t, \boldsymbol{X}_{\mathcal{G}_i}^t, \boldsymbol{X}_{\mathcal{G}_{-i}}^t$, and $\forall \, \tilde{\boldsymbol{D}}_{\mathcal{G}_i}^t, \tilde{\boldsymbol{D}}_{\mathcal{G}_{-i}}^t$, $\tilde{\boldsymbol{X}}_{\mathcal{G}_i}^t, \tilde{\boldsymbol{X}}_{\mathcal{G}_{-i}}^t$ if $F_i^t(\boldsymbol{X}_{\mathcal{G}_i}^t, \boldsymbol{D}_{\mathcal{G}_i}^t) = F_i^t(\tilde{\boldsymbol{X}}_{\mathcal{G}_i}^t, \tilde{\boldsymbol{D}}_{\mathcal{G}_i}^t)$, the following equality holds:

$$Y_i^t(D_i^t, \boldsymbol{D}_{\mathcal{G}_i}^t, \boldsymbol{D}_{\mathcal{G}_{-i}}^t) = Y_i^t(D_i^t, \tilde{\boldsymbol{D}}_{\mathcal{G}_i}^t, \tilde{\boldsymbol{D}}_{\mathcal{G}_{-i}}^t), \tag{1}$$

where $Y_i^t(D_i^t, \boldsymbol{D}_{\mathcal{G}_i}^t, \boldsymbol{D}_{\mathcal{G}_{-i}}^t)$ is the general case of unit $i$'s potential outcome. $\tilde{\boldsymbol{D}}_{\mathcal{G}_i}^t, \tilde{\boldsymbol{D}}_{\mathcal{G}_{-i}}^t, \tilde{\boldsymbol{X}}_{\mathcal{G}_i}^t, \tilde{\boldsymbol{X}}_{\mathcal{G}_{-i}}^t$ are the alternative assignments to $\boldsymbol{D}_{\mathcal{G}_i}^t, \boldsymbol{D}_{\mathcal{G}_{-i}}^t, \boldsymbol{X}_{\mathcal{G}_i}^t, \boldsymbol{X}_{\mathcal{G}_{-i}}^t$, respectively.

This assumption implies that once the response to the environment exposure function, i.e., $E_i^t$, is determined, the potential outcome of unit $i$ under treatment $D_i^t$ is fully specified. Then we extend the notation $Y_i(D_i)$ by incorporating the environment exposure $E_i^t$ and let $Y_i(D_i^t, E_i^t)$ denote unit $i$'s potential outcome under treatment $D_i^t$ and exposure $E_i^t$ at time step $t$.

Formally, we define our target estimand, the **C**onditional **A**verage **T**reatment **E**ffect with **I**nterference under **D**ynamic networks (**CATE-ID**), as an extension of the standard CATE to dynamic network settings with interference. Specifically, CATE-ID measures the expected difference in outcomes under alternative treatment assignments conditioning on the unit's covariates, historical information, and neighbors' covariates at time step $t$:

$$\tau_i^t = \mathbb{E}[Y_i^t(1, E_i^t = e_i^t)|\boldsymbol{x}_i^t, \mathcal{H}^t, \boldsymbol{X}_{\mathcal{G}_i}^t] - \mathbb{E}[Y_i^t(0, E_i^t = e_i^t)|\boldsymbol{x}_i^t, \mathcal{H}^t, \boldsymbol{X}_{\mathcal{G}_i}^t], \tag{2}$$

where $\mathcal{H}^t = \{\boldsymbol{X}^{<t}, \boldsymbol{D}^{<t}, \boldsymbol{A}^{<t}\}^2$ is the historical information that encodes past covariates, treatments, and network structures for time step $t$. CATE-ID incorporates environment exposure, enabling the assessment of an intervention's *intrinsic* causal effect on an individual while excluding interference. For instance, in the aforementioned study, we can capture the true effect of a mobility restriction itself on infection risk, eliminating the bias from the indirect effects transmitted through social contacts (e.g., a unit's infection risk may increase due to its neighbors' non-compliance with the mobility restriction, then the effect of the intervention may be underestimated).

## 3 Causal Identifiability

Building on the theoretical frameworks proposed in (Forastiere et al., 2021; Ma & Tresp, 2021), we introduce the assumptions adopted in our setting and formally establish the identifiability of CATE-ID defined in Eq. (2) for dynamic network environments with interference.

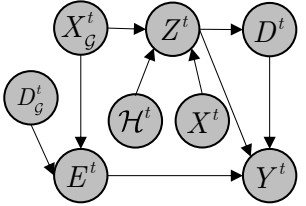

Figure 1: Causal Graph.

Previous literature (Shalit et al., 2017; Schwab et al., 2018; Ma et al., 2022) typically relies on the ignorability assumption, which presumes the absence of hidden confounders. In this work, we extend this assumption to allow for the presence of hidden confounders:

**Assumption 3.1. Extended Ignorability Assumption.** There exists a function $\Phi_z(\cdot)$: $\boldsymbol{z}_i^t = \Phi_z(\boldsymbol{x}_i^t, \boldsymbol{X}_{\mathcal{G}_i}^t, \mathcal{H}^t)$ that encodes the information of unit $i$'s covariates, historical factor, and its neighbors' covariates into *full confounders* $Z_i^t$, such that, given $Z_i^t$, the treatment $D_i^t$ and environment exposure $E_i^t$ are independent of the potential outcomes, i.e., $Y_i^t(1, E_i^t), Y_i^t(0, E_i^t) \perp D_i^t, E_i^t|Z_i^t$.

Assumption 3.1 implies that all confounders have been captured by $Z_i^t$. Noting that this assumption is conceptually aligned with the standard ignorability assumption which assumes that all confounders

---

[2]Following prior works in dynamic causality (Lim, 2018; Bica et al., 2020; Ma et al., 2021), past outcomes $\boldsymbol{Y}^{<t}$ are often excluded, as their influence is assumed to be captured by past covariates and treatments, with the goal of estimating marginal treatment effects at each time step rather than modeling outcome trajectories. Nonetheless, our framework can readily incorporate $\boldsymbol{Y}^{<t}$ if required by specific applications.

are measured in the observed covariates, our extended ignorability assumes that all confounders are absorbed into the learned latent factor $Z_i^t$. Furthermore, to guarantee correspondence between potential outcomes and observations, we generalize the standard consistency assumption (VanderWeele & Hernan, 2013) to the setting with environment exposure:

**Assumption 3.2. Consistency Assumption**. An individual's potential outcome under a particular treatment and environment exposure is exactly the outcome we would observe if the individual actually received that treatment and exposure, i.e., $Y_i^t(D_i^t, E_i^t) = Y_i^t$ with observed $D_i$ and $E_i^t$.

Based on the above definitions and assumptions, the causal graph for a single time step in the dynamic network setting is depicted in Figure 1, where the subscript denoting unit index is omitted for generality and simplicity. Formally, we establish the following theorem on the identifiability of CATE-ID in the presence of hidden confounders and interference:

**Theorem 3.3.** *If we can recover the distribution $p(Y_i^t|Z_i^t, E_i^t, D_i^t)$ and $p(Z_i^t|X_i^t, \mathcal{H}^t, \boldsymbol{X}_{\mathcal{G}_i}^t)$, the estimand CATE-ID presented in Eq.(2) can be identified from the observational dynamic network.*

*Proof.* The identified form of CATE-ID under interference is derived by:

$$
\begin{aligned}
\tau_i^t &= \mathbb{E}_Y[Y_i^t(1, E_i^t = e_i^t) - Y_i^t(0, E_i^t = e_i^t)|\boldsymbol{x}_i^t, \mathcal{H}^t, \boldsymbol{X}_{\mathcal{G}_i}^t] \\
&\overset{(1)}{=} \mathbb{E}_Z[\mathbb{E}_Y[Y_i^t(1, E_i^t = e_i^t) - Y_i^t(0, E_i^t = e_i^t)|\boldsymbol{x}_i^t, \mathcal{H}^t, \boldsymbol{X}_{\mathcal{G}_i}^t, Z_i^t]|\boldsymbol{x}_i^t, \mathcal{H}^t, \boldsymbol{X}_{\mathcal{G}_i}^t] \\
&\overset{(2)}{=} \mathbb{E}_Z[\mathbb{E}_Y[Y_i^t(1, E_i^t = e_i^t) - Y_i^t(0, E_i^t = e_i^t)|\boldsymbol{x}_i^t, \mathcal{H}^t, \boldsymbol{X}_{\mathcal{G}_i}^t, Z_i^t, E_i^t, D_i^t]|\boldsymbol{x}_i^t, \mathcal{H}^t, \boldsymbol{X}_{\mathcal{G}_i}^t] \quad (3) \\
&\overset{(3)}{=} \mathbb{E}_Z[\mathbb{E}_Y[Y_i^t(1, E_i^t = e_i^t) - Y_i^t(0, E_i^t = e_i^t)|Z_i^t, E_i^t, D_i^t]|\boldsymbol{x}_i^t, \mathcal{H}^t, \boldsymbol{X}_{\mathcal{G}_i}^t] \\
&\overset{(4)}{=} \mathbb{E}_Z[\mathbb{E}_Y[Y_i^t|Z_i^t, E_i^t = e_i^t, D_i^t = 1] - \mathbb{E}_Y[Y_i^t|Z_i^t, E_i^t = e_i^t, D_i^t = 0]|\boldsymbol{x}_i^t, \mathcal{H}^t, \boldsymbol{X}_{\mathcal{G}_i}^t],
\end{aligned}
$$

where equation $(1)$ is the straightforward expectation over full confounders $Z_i^t$, equation $(2)$ is derived by Assumption 3.1, equation $(3)$ can be inferred from condition independence such that $Y_i^t \perp \mathcal{H}^t, X_i^t, X_{\mathcal{G}_i}^t|Z_i^t, E_i^t$ from the causal graph, and equation $(4)$ is based on Assumption 3.2. □

## 4 METHODOLOGY

In this section, we introduce Dynamic SPillover modeling NETwork (DSPNET), a novel framework for estimating the target treatment effect estimand that explicitly addresses interference in dynamic network environments. The overall workflow of DSPNET in a single time step is illustrated in Figure 2, and its key components are detailed in the following subsections.

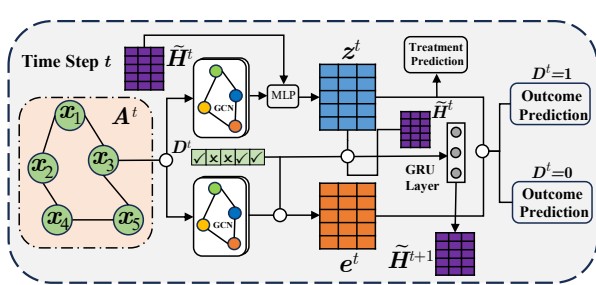

Figure 2: The workflow illustration of DSPNET.

### 4.1 REPRESENTATION LEARNING OF FULL CONFOUNDERS

As previously presented in Assumption 3.1, the full confounders $Z_i^t$ can be inferred via the function $\Phi_z(\cdot)$ by incorporating historical information, neighbor's covariates, and unit's own covariates. Here, we approximate the function $\Phi_z(\cdot)$ using multi-layer graph convolutional networks (GCNs) combined with multilayer perceptrons (MLPs) to learn the representation of the full confounders $Z_i^t$:

$$
\boldsymbol{z}_i^t = f_z^t([g_z^t(\boldsymbol{X}^t, \boldsymbol{A}^t)_i, \tilde{\boldsymbol{H}}_i^t]), \tag{4}
$$

where $g_z^t(\cdot)$ and $f_z^t(\cdot)$ represent the function parameterized by multiple GCN and MLP layers, $\tilde{\boldsymbol{H}}_i^t$ is the encoded historical state for unit $i$ at time step $t$. To model the historical state $\tilde{\boldsymbol{H}}_i^t$, we employ a recurrent neural unit, such as a Gated Recurrent Unit (GRU) (Cho et al., 2014) or a Long Short-Term Memory (LSTM) (Hochreiter & Schmidhuber, 1997), to encode full confounders, treatment assignment, and the historical state of the previous step:

$$
\tilde{\boldsymbol{H}}_i^t = \text{RNN}([\boldsymbol{z}_i^{t-1}, d_i^{t-1}], \tilde{\boldsymbol{H}}_i^{t-1}), \tag{5}
$$

where RNN$(\cdot)$ can be instantiated as either a GRU or an LSTM cell, depending on the implementation choice. The initial historical state, i.e., $\tilde{\boldsymbol{H}}_i^1$, is set to a zero vector.

## 4.2 INTERFERENCE MODELING

In many networks, interference depends not only on neighbors' treatment assignments, but also on how these treatments interact with intrinsic behavioral patterns of individuals. For example, a health intervention (e.g., jog) may affect a user only if its treated neighbors actively engage in and share health-related behaviors. However, previous works (Forastiere et al., 2021; Ma & Tresp, 2021) often model interference by simple aggregations (e.g., mean pooling), which oversimplify these patterns.

To more effectively model interference affected by neighbors' behavioral patterns, we propose inferring an interference representation $\boldsymbol{e}_i^t$ for each unit $i$ as a proxy for the environment exposure variable $E_i^t$, incorporating both the treatments and attributes of its neighbors for capturing the heterogeneous influences. Specifically, we first project each unit's observed covariates into a latent space via a function $g_r(\cdot)$, parameterized by a stack of GCN layers, to obtain the hidden state $\boldsymbol{r}_i^t$ of each unit $i$. Then we aggregate the hidden states of unit $i$'s neighbors by incorporating their treatment assignments to formulate the interference representation $\boldsymbol{e}_i^t$ for unit $i$ at time step $t$:

$$\boldsymbol{r}_i^t = g_r(\boldsymbol{X}^t, \boldsymbol{A}^t)_i, \quad \boldsymbol{e_i^t} = \sum_{j \in \mathcal{G}_i^t} d_j^t \cdot \boldsymbol{r}_j^t. \tag{6}$$

The resulting $\boldsymbol{e}_i^t$ can therefore be interpreted as a data-driven embedding of the influence with behaviors exerted by treated neighbors.

## 4.3 OUTCOME PREDICTION

Given the representation $\boldsymbol{z}_i^t$ of full confounders and the interference representation $\boldsymbol{e}_i^t$, our goal is to estimate the potential outcomes under alternative treatment assignments. To this end, we construct two separate MLP networks corresponding to treatment treatment $d = 1$ and $d = 0$, respectively. Formally, the potential outcome of unit $i$ w.r.t. treatment assignment $d_i^t$ at time step $t$ is given by:

$$f(\boldsymbol{z}_i^t, \boldsymbol{e}_i^t, d_i^t) = \begin{cases} \hat{y}_i^t(0) = f_0^t(\boldsymbol{z}_i^t, \boldsymbol{e}_i^t), & \text{if } d_i^t = 0 \\ \hat{y}_i^t(1) = f_1^t(\boldsymbol{z}_i^t, \boldsymbol{e}_i^t), & \text{if } d_i^t = 1 \end{cases}, \tag{7}$$

where the functions $f_0^t(\cdot)$ and $f_1^t(\cdot)$ are both parameterized by stacking multiple MLP layers, $\hat{y}_i^t(0)$ and $\hat{y}_i^t(1)$ represent the predicted potential outcomes under treatment $d_i^t = 0$ and $d_i^t = 1$, respectively. Then the predicted factual outcome is computed as $\hat{y}_i^t = (1 - d_i^t) \cdot \hat{y}_i^t(0) + d_i^t \cdot \hat{y}_i^t(1)$.

Given the predicted factual outcome $\hat{y}_i^t$ and observed outcome $y_i^t$ for each unit and time step, we employ the Mean Square Error (MSE) loss to minimize the discrepancy between them:

$$\mathcal{L}_y = \frac{1}{T} \sum_{t=1}^{T} \frac{1}{N} \sum_{i=1}^{N} (\hat{y}_i^t - y_i^t)^2. \tag{8}$$

## 4.4 BALANCING CONFOUNDER REPRESENTATIONS VIA GRADIENT REVERSAL

Estimating treatment effects requires addressing confounding bias, which arises from the differences in confounder distributions between treatment and control groups, and (Shalit et al., 2017) theoretically shows that learning representations that minimize distributional discrepancies between different groups can effectively reduce the upper bound of the estimation error. Motivated by this, we adopt an adversarial learning strategy with a Gradient Reversal Layer (GRL) (Ganin et al., 2016; Ma et al., 2021), which encourages confounder representations to be balanced across treatment groups, thus mitigating confounding bias while preserving predictive information.

First, we model treatment assignment with a prediction function $f_d(\cdot)$, parameterized by an MLP, which takes the full confounder representation $\boldsymbol{z}_i^t$ as input and outputs the probability of receiving treatment, i.e., $p(D_i^t = 1 | \boldsymbol{z}_i^t) = f_d(\boldsymbol{z}_i^t)$, and trained by the cross-entropy loss:

$$\mathcal{L}_d = -\frac{1}{T} \sum_{t=1}^{T} \frac{1}{N} \sum_{i=1}^{N} [d_i^t log(p(D_i^t = 1 | \boldsymbol{z}_i^t)) + (1 - d_i^t) log(1 - p(D_i^t = 1 | \boldsymbol{z}_i^t))]. \tag{9}$$

Then, the Gradient Reversal Layer operates as follows: let $\mathcal{L}$ denote the final loss function of the proposed DSPNET framework, which can be formulated as follows:

$$\mathcal{L} = \mathcal{L}_y + \alpha \mathcal{L}_d + \omega ||\boldsymbol{\Theta}||^2, \tag{10}$$

where $\boldsymbol{\Theta}$ denotes the set of learnable model parameters, $\alpha$ and $\omega$ are hyperparameters that control the contributions of the treatment prediction loss and the regularization term, respectively. Let $\boldsymbol{\Theta}_z$ represent the parameters associated with the full confounder representation learning module. Then, during backpropagation with GRL, the update for $\boldsymbol{\Theta}_z$ is modified as follows:

$$\boldsymbol{\Theta}_z = \boldsymbol{\Theta}_z - \eta(\frac{\partial \mathcal{L}_y}{\partial \boldsymbol{\Theta}_z} - \beta \frac{\partial \alpha \mathcal{L}_d}{\partial \boldsymbol{\Theta}_z} + \omega \frac{\partial ||\boldsymbol{\Theta}||^2}{\partial \boldsymbol{\Theta}_z}), \tag{11}$$

where $\eta$ is the learning rate. That is, when updating the parameters of $\boldsymbol{\Theta}_z$, we multiply the gradient from treatment prediction module by a negative constant $-\beta$ during backpropagation, and all other parameters $\boldsymbol{\Theta} \backslash \boldsymbol{\Theta}_z$ are updated by standard gradient descent. Intuitively, the GRL discourages the confounder representation from carrying predictive information to treatment, thereby aligning the representation distributions across treatment groups while preserving outcome-relevant information.

### 4.5 TIME AND SPACE COMPLEXITY ANALYSIS

At each time step $t$, DSPNET consists of four main components: $(i)$ an $L_g$-layer GCN backbone, $(ii)$ the interference module, $(iii)$ a GRU-based temporal encoder, and $(iv)$ an $L_m$-layer MLP for potential-outcome and treatment prediction.

For a graph with $N$ nodes, $M_t$ edges, and hidden dimension $d_h$, the per-layer GCN cost is $\mathcal{O}(Nd^2 + M_t d_h)$, leading to $\mathcal{O}(L_g(Nd_h^2 + M_t d_h))$ for the full backbone. The interference module performs sparse, edge-wise aggregation with cost $\mathcal{O}(M_t d_h)$. The GRU encoder updates node-level hidden states with matrix multiplications of size $d_h \times d_h$, giving $\mathcal{O}(Nd_h^2)$ per step. The MLP heads require $\mathcal{O}(L_m N d_h^2)$ across all nodes. Aggregating these components over a sequence of length $T$ and assuming $M_t = M$, the overall per-epoch time complexity is as follows:

$$\mathcal{O}(T((L_g + 1)Md_h + (L_g + 1 + L_m)Nd_h^2)) = \mathcal{O}(T(L_g Md_h + (L_g + L_m)Nd_h^2)),$$

which is linear in the number of edges $M$ and nodes $N$. The space complexity of DSPNET is dominated by storing the sparse adjacency structure and node embeddings at each time step, i.e., $\mathcal{O}(M_t + Nd_h)$, which is also linear in $M$ and $N$.

## 5 EXPERIMENTS

### 5.1 DATASETS

**Flickr**: Flickr (Tang & Liu, 2011) is a popular image and video-based social network, where users connect and share multimedia content. In this dataset, each node corresponds to a user, and edges represent friendships between users. The features of a node are constructed using bag-of-words representations of user's interest tag.

**BlogCatalog**: BlogCatalog (Zafarani & Liu, 2009) is a social networking platform where bloggers share their blogs and interact with each other. In the BlogCatalog dataset, each node represents a blogger, and edges indicate social relationships between bloggers. The features are also constructed using bag-of-words representations of the bloggers' posts.

To construct a dynamic network, we introduce temporal variations by randomly adding or removing $p\%$ of edges in the underlying network structure and apply Gaussian noise perturbations to the same proportion of covariates in each time step (25 time steps in total) to simulate real-world fluctuations. Using the generated dynamic network, we simulate confounders, treatment assignments, and interference-aware potential outcomes via an auto-regressive process at each time step, the detailed data generation procedure can be found in Appendix A.

### 5.2 BASELINES

Here we briefly introduce the baseline methods for estimating treatment effects in our evaluation, which can be categorized into two categories:

Table 1: CATE-ID performance comparison by varying degrees of network dynamics. **Bold**: the best results. Underline: the 2nd best results. Lower is better. $^\dagger$ indicates statistically significant improvement over the strongest baseline (t-test, $p$-value $< 0.05$).

| Datasets | Methods | $p\% = 0.1\%$ | | $p\% = 0.5\%$ | | $p\% = 1.0\%$ | |
|---|---|---|---|---|---|---|---|
| | | $\sqrt{\epsilon_{PEHE}}$ | $\epsilon_{ATE}$ | $\sqrt{\epsilon_{PEHE}}$ | $\epsilon_{ATE}$ | $\sqrt{\epsilon_{PEHE}}$ | $\epsilon_{ATE}$ |
| Flickr | DESCN | 17.982 ±3.872 | 16.996 ±3.821 | 29.551 ±9.891 | 28.273 ±9.605 | 30.396 ±6.052 | 29.058 ±5.873 |
| | DFITE | 17.404 ±2.933 | 3.083 ±0.486 | 20.527 ±3.514 | 3.306 ±0.410 | 22.459 ±6.097 | 3.545 ±0.772 |
| | DERCFR | 21.704 ±3.786 | 17.246 ±3.825 | 31.406 ±5.804 | 23.916 ±5.023 | 31.847 ±6.121 | 25.925 ±5.850 |
| | CFR | 24.218 ±3.939 | 2.754 ±0.599 | 26.716 ±3.995 | 2.841 ±0.444 | 29.372 ±6.836 | 3.185 ±0.632 |
| | NetEST | 6.822 ±1.107 | 1.405 ±0.219 | 10.708 ±2.935 | 2.575 ±0.386 | 11.000 ±1.806 | 2.418 ±0.421 |
| | Deconfounder | 8.338 ±1.230 | 4.738 ±0.844 | 11.724 ±3.344 | 6.854 ±2.175 | 12.249 ±1.912 | 7.132 ±1.399 |
| | SPNET | 8.693 ±1.030 | 1.204 ±0.216 | 11.320 ±2.769 | 1.397 ±0.340 | 11.797 ±1.679 | 1.542 ±0.384 |
| | DNDC | 2.589 ±0.959 | 1.618 ±0.781 | 3.062 ±0.379 | 1.915 ±0.187 | 3.291 ±0.522 | 2.194 ±0.578 |
| | **DSPNET** | **1.497 ±0.145**$^\dagger$ | **0.890 ±0.080**$^\dagger$ | **2.062 ±0.498**$^\dagger$ | **1.144 ±0.128**$^\dagger$ | **2.189 ±0.205**$^\dagger$ | **1.351 ±0.209**$^\dagger$ |
| BlogCatalog | DESCN | 23.430 ±3.422 | 22.348 ±3.428 | 26.393 ±4.581 | 25.198 ±4.429 | 28.458 ±4.822 | 27.229 ±4.654 |
| | DFITE | 11.841 ±3.243 | 3.446 ±0.427 | 14.028 ±4.200 | 3.618 ±0.786 | 14.483 ±3.005 | 3.559 ±0.599 |
| | DERCFR | 35.321 ±8.824 | 24.921 ±3.295 | 39.149 ±5.028 | 29.219 ±4.523 | 39.286 ±7.888 | 30.360 ±8.279 |
| | CFR | 11.547 ±3.164 | 1.295 ±0.249 | 13.935 ±4.166 | 1.171 ±0.168 | 14.546 ±3.352 | 1.279 ±0.246 |
| | NetEST | 8.539 ±1.074 | 1.586 ±0.218 | 9.871 ±1.161 | 1.847 ±0.204 | 9.533 ±1.255 | 1.835 ±0.223 |
| | Deconfounder | 13.067 ±1.863 | 8.884 ±1.170 | 14.870 ±2.515 | 9.709 ±1.711 | 15.037 ±2.336 | 9.910 ±1.491 |
| | SPNET | 9.569 ±1.742 | 2.298 ±0.859 | 10.681 ±1.771 | 2.597 ±0.705 | 10.288 ±1.736 | 2.149 ±0.522 |
| | DNDC | 2.475 ±0.462 | 1.454 ±0.400 | 2.419 ±0.332 | 1.319 ±0.329 | 3.367 ±0.757 | 1.723 ±0.530 |
| | **DSPNET** | **1.464 ±0.119**$^\dagger$ | **0.845 ±0.105**$^\dagger$ | **1.506 ±0.237**$^\dagger$ | **0.913 ±0.204**$^\dagger$ | **2.227 ±0.378**$^\dagger$ | **1.183 ±0.290**$^\dagger$ |

**Non-Networked**: (1) CFR (Shalit et al., 2017) is a deep learning-based approach which mitigates distributional imbalance between treatment and control groups by incorporating the Wasserstein distance regularizer; (2) DESCN (Zhong et al., 2022) is a model for estimating treatment effects by capturing integrated information on treatment propensity, response, and hidden treatment effects through a cross-network in a multi-task learning framework; (3) DFITE (Wang et al.) leverages diffusion models to capture the latent space of these unobserved confounders by modeling the reverse diffusion process as a Markov chain to estimate treatment effects; (4) DERCFR (Wu et al., 2022) is a framework which identifies and separates confounders from non-confounders for reducing bias of treatment effect estimation.

**Networked**: (5) NetEST (Jiang & Sun, 2022) formulates the treatment effect estimation problem as a multi-task learning task, employing representation learning techniques to align the distributions of treated and control groups for networked environment; (6) Deconfounder (Guo et al., 2020c) utilize graph neural network to capture the hidden confounders by leveraging the network structure for treatment effect estimation; (7) DNDC (Ma et al., 2021) is designed to leverage current and historical networked observational data to learn representations of hidden confounders over time for the treatment effect estimation in dynamic network environment; (8) SPNet (Huang et al., 2023) aims to model the interference by developing a attention mechanism for treatment effect estimation in static networked environment.

## 5.3 EXPERIMENTAL SETUP

For each dataset, we run each experiment ten times and report the average performance. In each run, the dataset is randomly split into training-60%, validation-20%, and test-20% set. As described in data generation process in Appendix A, we set the degree of dynamic $p\% = 0.1\%$, and the strength of interference $C = 50$ unless otherwise specified. We adopt the grid search strategy based on the validation performance to identify the optimal hyperparameter configuration. For hyperparameters of DSPNET, the learning rate $\eta$ is set to $4 \times 10^{-3}$, $\alpha$ and $\beta$ range in $\{1,2,3,4\}$, and $\omega$ ranges in $\{10^{-1}, 10^{-2}, 10^{-3}, 10^{-4}\}$. We use Adam (Kingma & Ba, 2014) as the model optimizer and GRU cell to model the historical state. For baselines designed for static data, we train a separate model at each time step and report the averaged performance across all steps. More detailed experiments such as using LSTM cell, different strength of historical influence, different influence of network structure and different balancing strategies can be found in Appendix C.

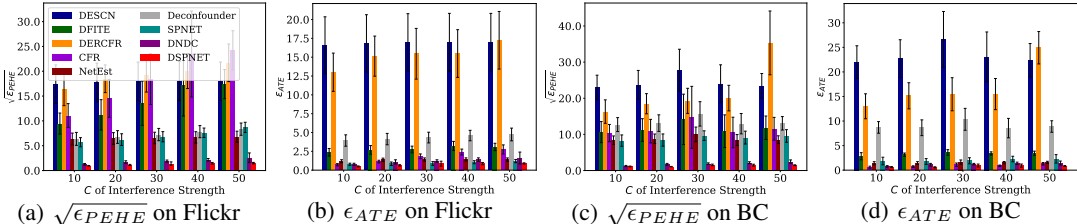

Figure 3: Performance under varying strengths of network interference, "BC" denotes BlogCatalog.

## 5.4 Direct Evaluation of CATE-ID Estimation

In this section, we directly evaluate the performance of CATE-ID estimation using the following two commonly used metrics: rooted precision in estimation of heterogeneous effect $\sqrt{\epsilon_{PEHE}}$ and mean absolute error on average treatment effect $\epsilon_{ATE}$, where $\sqrt{\epsilon_{PEHE}}$ and $\epsilon_{ATE}$ aim to measure the accuracy of unit-level and population-level treatment effect estimation across $T$ time steps:

$$\sqrt{\epsilon_{PEHE}} = \frac{1}{T}\sum_{t=1}^{T}\sqrt{\frac{1}{N}\sum_{i=1}^{N}(\tau_i^t - \hat{\tau}_i^t)^2}, \quad \epsilon_{ATE} = \frac{1}{T}\sum_{t=1}^{T}|\frac{1}{N}\sum_{i=1}^{N}\tau_i^t - \frac{1}{N}\sum_{i=1}^{N}\hat{\tau}_i^t|, \quad (12)$$

where $\tau_i^t = y_i^t(1) - y_i^t(0)$ and $\hat{\tau}_i^t = \hat{y}_i^t(1) - \hat{y}_i^t(0)$ are the ground-truth and predicted CATE-ID, respectively. We take the average of the two metrics over $T$ time steps for the final results.

### 5.4.1 Comparison under Varying Degrees of Network Dynamics

First, we evaluate the performance of different models in estimating treatment effects under varying degrees of network dynamics. Specifically, we set $p\% = \{0.1\%, 0.5\%, 1.0\%\}$ to control the level of structural and feature perturbations, and compare the CATE-ID estimation performance of all models across these settings. The experimental results are reported in Table 1.

As shown in the results, the proposed DSPNET consistently achieves the best performance across all dynamic settings. Notably, DSPNET maintains stable performance as the proportion of dynamic edge and feature perturbations increases, demonstrating its robustness to network dynamics. The poor performance of non-network baselines (i.e., CFR, DERCFR, DESCN, DFITE) highlights that ignoring network dependencies leads to biased treatment effect estimation in networked environments. Although NetEST, Deconfounder, and SPNET incorporate network structures, they assume a static network setting, resulting in performance degradation under dynamic conditions. DNDC, which is tailored for dynamic networks, achieves relatively strong results but does not explicitly model interference. Consequently, its performance remains consistently inferior to DSPNET, underscoring the importance of capturing spillover effects for accurate treatment effect estimation.

### 5.4.2 Comparison under Varying Strengths of Network Interference

Then, we investigate how model performance varies under different strengths of network interference. Specifically, we keep all other parameters as default during the data generation process and control the strength of network interference using $C = \{10, 20, 30, 40, 50\}$. The performance of different models under different interference strength is illustrated in Figure 3.

The results show that the proposed DSPNET consistently outperforms all other models across all levels of network interference, and its performance remains relatively stable even under high interference strengths (e.g., $C = 40$ or $50$), demonstrating its effectiveness in handling complex spillover effects and its suitability for real-world applications. While SPNET is designed to model spillover effects, it does not account for temporal dynamics, resulting in inferior performance compared to dynamic models such as DNDC and DSPNET. Moreover, as $C$ increases, the performance gap between DNDC and DSPNET becomes more pronounced, further highlighting the importance of explicitly modeling network interference in treatment effect estimation.

### 5.4.3 ABLATION STUDY

To investigate the contribution of each component in our proposed DSPNET model, we conduct an ablation study by evaluating the following model variants:

($i$) *w/o GRL*: This variant removes the Gradient Reversal Layer which is responsible for balancing the confounder representation across treatment groups and mitigating the confounding bias.

($ii$) *w/o IM*: This variant eliminates the Interference Modeling component, thereby excluding the learning of interference representation for outcome estimation.

($iii$) *w/o GRU*: This variant excludes the Gated Recurrent Unit (GRU), disabling the model's ability to capture temporal dependencies and historical information.

Table 2: Ablation Study.

| Variants | Flickr | | Blogcatalog | |
|---|---|---|---|---|
| | $\sqrt{\epsilon_{PEHE}}$ | $\epsilon_{ATE}$ | $\sqrt{\epsilon_{PEHE}}$ | $\epsilon_{ATE}$ |
| Original | **1.497** ±0.145 | **0.890** ±0.080 | **1.464** ±0.119 | **0.845** ±0.105 |
| *w/o GRL* | 2.179 ±0.266 | 0.986 ±0.108 | 1.886 ±0.227 | 1.089 ±0.247 |
| *w/o IM* | 1.938 ±0.242 | 1.245 ±0.203 | 1.822 ±0.138 | 1.118 ±0.164 |
| *w/o GRU* | 10.235 ±1.768 | 6.854 ±1.014 | 10.652 ±0.718 | 3.547 ±0.912 |

The results are shown in Table 2. Specifically, removing the GRL component results in a moderate drop, underscoring its role in mitigating confounding bias. Excluding interference modeling leads to further degradation, highlighting the importance of capturing spillover effects in networked environments. The most substantial performance loss is observed when removing GRU component, demonstrating the critical role of capturing historical information in dynamic settings. Overall, these findings validate the necessity of each component in ensuring the effectiveness of DSPNET.

### 5.4.4 HYPERPARAMETER ANALYSIS

To analyze the impact of hyperparameters $\alpha$ (which controls the contribution of treatment prediction) and $\beta$ (which regulates the gradient reversal layer), we conduct a sensitivity analysis by varying both parameters within the set $\{1, 2, 3, 4\}$. We visualize the performance trends across different combinations of $\alpha$ and $\beta$ using a histogram to facilitate comparison. As shown in Figure 4, the proposed DSPNET exhibits stable performance across different hyperparameter configurations, indicating that it is not highly sensitive to specific choices of $\alpha$ and $\beta$. Notably, when $\alpha, \beta \in \{1, 2\}$, DSPNET attains relatively better performance, suggesting that balanced and moderate contributions from treatment prediction and gradient reversal yield better results.

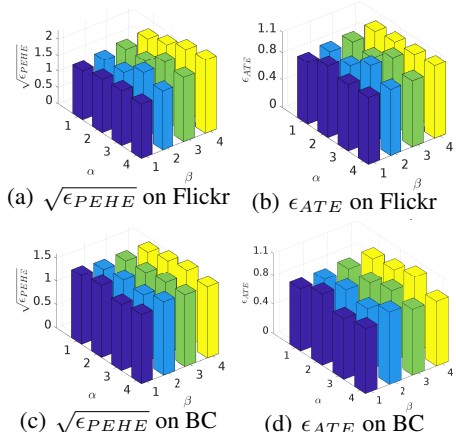

(a) $\sqrt{\epsilon_{PEHE}}$ on Flickr  (b) $\epsilon_{ATE}$ on Flickr

(c) $\sqrt{\epsilon_{PEHE}}$ on BC  (d) $\epsilon_{ATE}$ on BC

Figure 4: Hyperparameter Analysis.

### 5.5 EVALUATING TREATMENT PRIORITIZATION RULES

Beyond directly measuring the CATE-ID estimation accuracy by $\sqrt{\epsilon_{PEHE}}$ and $\epsilon_{ATE}$ with ground-truth CATE-ID, we further evaluate the quality of treatment prioritization rules induced by the estimated CATE-ID of different estimators by adopting **Rank-weighted Average Treatment Effect (RATE)** (Yadlowsky et al., 2025). RATE provides a principled approach to assess how good the estimated CATE-ID is without requiring access to the ground-truth CATE-

Table 3: RATE results, higher is better.

| Methods | Flickr | | BlogCatalog | |
|---|---|---|---|---|
| | $R_{AUTOC}$ | $R_{QINI}$ | $R_{AUTOC}$ | $R_{QINI}$ |
| Deconfounder | 0.03±0.39 | 0.01±0.13 | 0.19 ± 0.14 | 0.07 ± 0.06 |
| SPNET | 0.26±0.24 | 0.10±0.07 | 0.05 ± 0.17 | 0.02 ± 0.06 |
| NetEST | 0.51±0.36 | 0.15±0.12 | 0.35 ± 0.15 | 0.15 ± 0.04 |
| DNDC | 2.72±0.69 | 1.04±0.27 | 3.03 ± 0.68 | 1.09 ± 0.47 |
| **DSPNET** | **2.98±0.64** | **1.13±0.25** | **3.91 ± 0.56** | **1.52 ± 0.22** |

ID values, focusing on its treatment prioritization ability, i.e., the extent to which individuals with higher estimated CATE-ID truly benefit more from treatment, it provides a complementary perspective on the quality of estimated CATE-ID beyond error-based metrics. The formal definition and more details about RATE metric can be found in the Appendix B.

We report the $R_{AUTOC}$ and $R_{QINI}$ scores—two variants of RATE—averaged over all time steps in Table 3, comparing the proposed DSPNET with network-based baselines (Deconfounder, SPNET,

NetEst, and DNDC) on BlogCatalog and Flickr with default generation. Results for those non-network baselines are omitted, as their performance was consistently poor in our earlier evaluations. As shown, DSPNET achieves higher and more stable RATE scores across datasets, demonstrating superior treatment prioritization capability over the other network-based baseline methods.

### 5.6 TIME AND SPACE OVERHEAD

To further evaluate the time and space overhead of DSPNET under large-scale dynamic networks, we scale the number of nodes and edges to roughly 1×, 2×, 4×, 8× and 10× of the original size on Flickr and BlogCatalog. DSPNET was trained on each scaled dataset under the same hardware configuration, and we report the wall-clock time per epoch and peak GPU memory usage as shown in Table 4.

Table 4: Time and Space Overhead

| Dataset | Scale | #Nodes | #Edges | Time | GPU Mem |
|---|---|---|---|---|---|
| Flickr | 1× | 7.5K | 239K | 0.24s | 2.8GB |
| | 2× | 15K | 478K | 0.31s | 5.6GB |
| | 4× | 30K | 956K | 0.51s | 11.3GB |
| | 8× | 60K | 1.91M | 0.94s | 22.5GB |
| | 10× | 75K | 2.39M | 1.14s | 28.1GB |
| BlogCatalog | 1× | 5K | 171K | 0.24s | 1.7GB |
| | 2× | 10K | 342K | 0.29s | 3.4GB |
| | 4× | 20K | 684K | 0.39s | 6.8GB |
| | 8× | 40K | 1.36M | 0.65s | 13.2GB |
| | 10× | 50K | 1.71M | 0.81s | 16.5GB |

Overall, the results show that both runtime and memory scale approximately linearly with graph size, consistent with our theoretical complexity analysis. Moreover, DSPNET continues to train and infer reliably even on graphs with 75k nodes and 2.39 million edges, suggesting that the framework is capable of handling substantially larger dynamic networks in practice.

## 6 RELATED WORKS

treatment effect estimation from observational data has received considerable attention in recent years, leading to the development of numerous methodological approaches. Unlike traditional methods (Shalit et al., 2017; Zhong et al., 2022; Wang et al.; Wu et al., 2022; Yao et al., 2018) that assume independent and identically distributed (i.i.d.) samples, networked data violate this assumption, as an individual's outcome can be influenced by their neighbors. To address confounding in such settings, several approaches—such as Deconfounder (Guo et al., 2020c), NetEST (Jiang & Sun, 2022), CONE (Guo et al., 2020b), and IGNITE (Guo et al., 2021)—have been proposed. These methods leverage network structures to capture hidden confounders and adopt various balancing strategies to mitigate confounding bias for reliable treatment effect estimation. However, they do not explicitly model network interference. To address interference, LCVA (Rakesh et al., 2018) utilizes a variational autoencoder to capture spillover effects between units. SPNET (Huang et al., 2023) further refines this by modeling heterogeneous spillover magnitudes across neighbor pairs. Additionally, (Ma & Tresp, 2021) explores interference modeling via simple aggregation of neighbors' treatments, while HyperSCI (Ma et al., 2022) extends this idea to hypergraphs using attention mechanisms based on neighbors' representations. However, these methods are designed for static networks and cannot be directly applied to dynamic environments. Although DNDC (Ma et al., 2021) aims to estimate treatment effects in dynamic graphs, it does not account for interference. In contrast, our work targets on dynamic network environments, while simultaneously modeling hidden confounders and interference, thereby addressing both time-evolving confounders and spillover effects.

## 7 CONCLUSION

In this paper, we study the problem of treatment effect estimation in dynamic network environments, explicitly accounting for time-varying hidden confounders and network interference. We begin by introducing a new treatment effect estimand CATE-ID tailored for dynamic settings with interference and formally prove its identifiability. Building on this theoretical foundation, we propose DSPNET, a novel framework that leverages both the evolving network structure and historical information to model dynamic hidden confounders and interference, and then learns the representations of confounders and environment exposure to enable accurate treatment effect estimation over time. Extensive experiments demonstrate the superiority of our framework over existing methods for estimating treatment effects from dynamic networked observational data.

## 8 REPRODUCIBILITY STATEMENT

We have made significant efforts to ensure the reproducibility of our work. For the proposed algorithm, we provide an anonymous link to the full source code. The assumptions required for Theorem 3.3 as well as its complete proof are clearly presented in Section 3. For experimental evaluation, we detail the entire dataset generation and processing procedure in Appendix A, and provide additional experimental results and analysis in Appendix C. Together, these resources allow researchers to fully reproduce our theoretical results and empirical findings.

## 9 ACKNOWLEDGEMENTS

The authors thank the anonymous reviewers for their time and thoughtful comments.

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

APPENDIX

## A  DATA GENERATION

In this section, we introduce how to generate the treatment, confounders, interference, and potential outcomes for dataset BlogCatalog and Flickr:

**Step 1**: Simulate the full confounders $z_i^t$. we first introduce how to leverage the historical information and network structure to simulate the full confounders. The full confounders $z_i^t$ for unit $i$ at time step $t$ could be simulated as follows:

$$z_i^t = \frac{1}{\lambda_1 + \lambda_2 + \lambda_3}(\lambda_1 \boldsymbol{\Psi}_i^t + \lambda_2 \sum_{u \in \mathcal{G}_i^t} f_L(\boldsymbol{x}_u^t) + \lambda_3 f_L(\boldsymbol{x}_i^t) + \boldsymbol{\epsilon}^t), \tag{13}$$

where function $f_L(\cdot)$ maps the features in terms of bag-of-words to topic distribution by training a LDA topic model with 50 topics (Pritchard et al., 2000) such that the dimension of the output of $f_L(\cdot)$ is 50, thus $z_i^t \in \mathbb{R}^{50}$, $\boldsymbol{\Psi}_i^t \in \mathbb{R}^{50}$ represents the historical information, $\mathcal{G}_i^t$ denotes the neighbor set of unit $i$ at time step $t$. $\lambda_1, \lambda_2$ and $\lambda_3$ are to control the influence of historical information, network structure and current unit's covariates to the hidden confounders, respectively. $\lambda_1, \lambda_2, \lambda_3$ are set to $10, 3, 3$ by default; $\boldsymbol{\epsilon}^t \in \mathbb{R}^{50}$ is the noise vector and each element is sampled from the normal distribution $\epsilon_j^t \sim \mathcal{N}(0, 0.001)$; Each element $\Psi_{i,j}^t$ can be obtained using $q$-order autoregressive (Mills, 1990) as follows:

$$\Psi_{i,j}^t = \frac{1}{q}(\sum_{r=1}^{q} \rho_{r,j} z_{i,j}^{t-r} + \sum_{r=1}^{q} \varrho_r d_i^{t-r}), \tag{14}$$

where $z_{i,j}^{t-r}$ denotes the $j$-th element of confounders $z_i^{t-r}$ at time step $t-r$, $d_i^{t-r}$ represents the treatment assignment of unit $i$ at time step $t-r$, $\rho_{r,j} \sim \mathcal{N}(1 - r/q, 1/q)$ and $\varrho_r \sim \mathcal{N}(0, 0.02)$ are to control the impact of historical confounders and treatment assignments from previous $q$ time steps, here we follow (Ma et al., 2021) and set the order $q = 3$. Noting that $\boldsymbol{\Psi}_i^t$ at initial time step $t = 1$ is set to zero vector.

**Step 2**: Simulate observed treatment assignment $d_i^t$. First we randomly select two vectors $r_0^t, r_1^t$ from the current hidden confounders $\boldsymbol{Z}^t$ in the LDA topic space as the centroids for treatment and control group. The similarity between a unit's confounder representation $z_i^t$ and each centroid is measured by their inner product:

$$s_{i,0}^t = r_0^{t^T} z_i^t, s_{i,1}^t = r_1^{t^T} z_i^t, \tag{15}$$

which indicates how close unit $i$ is to the control or treatment centroid in the topic space, $r_0^{t^T}$ represents the transpose of $r_0^t$. Then we can simulate the treatment assignment $d_i^t$ for unit $i$ at time step $t$ as follows:

$$d_i^t \sim \mathcal{B}(\frac{exp(s_{i,1}^t)}{exp(s_{i,1}^t) + exp(s_{i,0}^t)}), \tag{16}$$

where $\mathcal{B}$ denotes the Bernoulli distribution.

**Step 3**: Simulate interference (spillover effect) $\mathcal{I}_i^t$. The interference $\mathcal{I}_i^t$ of unit $i$'s neighbors on it can be computed as follows:

$$\mathcal{I}_i^t = C \cdot \sum_{j \in \mathcal{G}_i^t} \boldsymbol{w}_p^{t^T} f_L(\boldsymbol{x}_j) \cdot d_j^t, \tag{17}$$

where $C$ is the scaling value to control the strength of interference in the network, $\boldsymbol{w}_p^t$ is the weight vector sampled from uniform distribution $\mathcal{N}(\boldsymbol{0}, \boldsymbol{I}_{50})$ where $\boldsymbol{I}_{50}$ is the identity matrix, the inner product $\boldsymbol{w}_p^{t^T} f_L(\boldsymbol{x}_j)$ quantifies neighbor $j$'s influence strength on unit $i$, which contributes to the interference if the neighbor receives treatment.

**Step 4**: Simulate potential outcomes. Given the aforementioned steps, the potential outcomes could be formulated as follows:

$$\begin{aligned} y_i^t(1) &= \kappa(s_{i,0}^t + s_{i,1}^t) + \mathcal{I}_i^t + \epsilon_y^t, \\ y_i^t(0) &= \kappa s_{i,0}^t + \mathcal{I}_i^t + \epsilon_y^t, \end{aligned} \tag{18}$$

Table 5: Statistical Summaries on Flickr and BlogCatalog

| Dataset | Flickr | BlogCatalog |
|---|---|---|
| Time Steps | 25 | 25 |
| # Units | 7575 | 5196 |
| # Averaged Edges | 239623 | 171661 |
| # Features | 12047 | 8189 |
| % Treated Ratio | $49.48 \pm 2.07$ | $49.53 \pm 2.00$ |
| Average ATE | $16.19 \pm 12.87$ | $22.00 \pm 17.10$ |

where $\kappa$ is the scaling factor and set to 100 by default, $\epsilon_y^t \sim \mathcal{N}(0, 1)$ is the sampled Gaussian noise. Then the observed outcome for unit $i$ at time step $t$ could be obtained by:

$$y_i^t = d_i^t \cdot y_i^t(1) + (1 - d_i^t) \cdot y_i^t(0). \tag{19}$$

Then, based on the above data generation process and default parameter settings, we randomly run the simulations 10 times on the Flickr and BlogCatalog datasets. The statistical summaries of the generated data are presented in Table 5.

## B  RANK-WEIGHTED AVERAGE TREATMENT EFFECT (RATE)

In practical applications of treatment effect estimation, the primary goal often lies not only in estimating the conditional average treatment effect (CATE) for each individual, but also in constructing treatment prioritization rules that rank individuals according to their estimated treatment effects. To evaluate the quality of such prioritization rules, (Yadlowsky et al., 2025) proposed the Rank-weighted Average Treatment Effect (RATE), a general evaluation framework that quantifies how well an estimated prioritization rule identifies those who truly benefit most from treatment. At a high level, the RATE captures the extent to which individuals who are highly ranked by the prioritization rule are more responsive to treatment than randomly selected individuals, which is another perspective to evaluate the quality of the estimated CATE.

Unlike error-based metrics such as $\sqrt{\epsilon_{PEHE}}$ and $\epsilon_{ATE}$, RATE does not require access to the true CATE values for each individual. Instead, it leverages the ranking induced by estimated CATEs, thereby providing a more practical evaluation criterion in real-world settings where the true individual treatment effect is unobservable.

### B.1  FORMAL DEFINITION OF RATE

Let $S(X_i)$ denote a treatment prioritization score assigned to the individual $X_i$, where $S(\cdot)$ is a priority scoring function that could be a learned priority rule, a learned CATE estimate, a hardcoded heuristic, or something else. In this work, our goal is to estimate the quality of the estimated CATE-ID, we take the estimated CATE-ID for each individual as the prioritization score:

$$S(X_i) = \hat{\tau}(X_i), \tag{20}$$

a larger value of $S(X_i)$ implies that the individual should be treated first. For a given quantile level $0 < q \leq 1$, we define the **Targeting Operator Characteristic (TOC)** as:

$$\text{TOC}(q; S) = \mathbb{E}[Y(1) - Y(0) \mid F(S(X_i)) \geq 1 - q] - \mathbb{E}[Y(1) - Y(0)], \tag{21}$$

where $F(S(X))$ is the cumulative distribution function of the prioritization score $S(X_i)$. Intuitively, $\text{TOC}(q; S)$ measures the incremental gain in treatment effect when prioritizing the top-$q$ fraction of individuals based on the priority scoring function $S(\cdot)$, compared to the overall population. Noting that if $q = 1$, the first term in Eq.(21) is the average treatment effect, and so $\text{TOC}(q; S) = 0$.

After defining the $\text{TOC}(q; S)$, we can involve evaluating prioritization rules (based on the estimated CATE-ID) in terms of weighted averages of the TOC. Formally, **Rank-weighted Average Treatment Effect (RATE)** is defined as the a weighted integral of the TOC curve:

$$\text{RATE}(w; S) = \int_0^1 \text{TOC}(q; S) \cdot w(q) dq, \tag{22}$$

where $w(q)$ is a nonnegative weight function that emphasizes different regions of the ranking. Different choices of $w(q)$ yield different variants of RATE. Here we introduce two variants of RATE based on different weight function as follows

(1) **Area Under the Targeting Operator Characteristic (AUTOC)**. When the weight function is chosen as uniform, i.e., $w(q) = 1$, RATE reduces to the the area under the TOC curve (AUTOC):

$$R_{AUTOC} = \int_0^1 \text{TOC}(q; S) dq. \tag{23}$$

$R_{AUTOC}$ provides an overall summary of how much benefit the prioritization rule delivers across the entire population distribution. A higher $R_{AUTOC}$ indicates that the ranking consistently selects individuals with higher treatment benefit, demonstrating higher quality of the treatment prioritization rule induced by the estimated CATE-ID.

(2) **Qini coefficient**. The Qini coefficient is a widely used variant of RATE that emphasizes the ability of a treatment prioritization rule to correctly identify individuals with the largest treatment effects. Its definition can be written as:

$$R_{QINI} = \int_0^1 q \cdot \text{TOC}(q; S) dq, \tag{24}$$

we can see that it uses the quantile level $q$ as the weight function. Compared to AUTOC, which applies uniform weighting over the entire population, the Qini coefficient assigns more weight to the upper quantiles of the prioritization rule, thereby stressing the importance of ranking performance among the top-scoring individuals.

## C    ADDITIONAL EXPERIMENTS

### C.1    MODEL ARCHITECTURE DETAILS

To enhance the transparency and reproducibility of our method, we provide a detailed description of the default model architecture used in the experiments. Table 6 summarizes the configuration of the backbone components, including the number of GCN and MLP layers, the hidden dimensions of GCN, MLP and recurrent layers, activation functions, and other relevant hyperparameters about training details.

Table 6: Model architecture specifications and training details.

| Component | Layers | Hidden Dimensions | Activation | Remarks |
|---|---|---|---|---|
| GCN Backbone | 1 | 70 | ReLU | - |
| MLP Backbone | 2 | 70 | ReLU | - |
| Recurrent Backbone | - | 70 | - | Gated Recurrent Unit |
| Treatment-MLP | 2 | 100 | Sigmoid | MLP for treatment |
| Dimension of Confounders | – | 70 | – | Final representation of confounders |
| Dimension of Interference | – | 70 | – | Final representation of exposure |
| Optimizer | – | – | – | Adam, learning rate = $4e-3$, weight decay = $1e-2$ |
| Other Hyperparameters | – | – | – | epochs=1000, $\alpha = 1$, $\beta = 1$, $\omega = 1e-4$ |

### C.2    PERFORMANCE COMPARISON UNDER VARYING INFLUENCE OF HISTORY

In this section, we investigate the impact of different intensities of historical information on the model's performance in estimating treatment effects. Specifically, we keep all other parameters in the data generation process as default and vary the value of $\lambda_1 = \{5,10,20\}$ to examine how different levels of historical influence affect the model's performance. The results are presented in Table 7.

We can see that our proposed DSPNET model consistently outperforms all baselines across different levels of historical influence. Most baseline methods exhibit significant deteriorating performance as $\lambda_1$ increases, which suggests that stronger historical dependencies make it more challenging for these models to accurately estimate treatment effects. However, our proposed model DSPNET suffers the least, demonstrating its superiority on capturing time-varying hidden confounder and interference on dynamic network environment.

Table 7: CATE-ID performance comparison by varying influence of historical information on Flickr and BlogCatalog datasets. **Bold**: the best results. Underline: the 2nd best results. Lower is better. $^{\dagger}$ indicates statistically significant improvement over the strongest baseline (t-test, $p$-value $< 0.05$).

| Datasets | Methods | $\lambda_1 = 5$ | | $\lambda_1 = 10$ | | $\lambda_1 = 20$ | |
|---|---|---|---|---|---|---|---|
| | | $\sqrt{\epsilon_{PEHE}}$ | $\epsilon_{ATE}$ | $\sqrt{\epsilon_{PEHE}}$ | $\epsilon_{ATE}$ | $\sqrt{\epsilon_{PEHE}}$ | $\epsilon_{ATE}$ |
| Flickr | DESCN | 10.637 ±2.198 | 9.621 ±2.162 | 17.982 ±3.872 | 16.996 ±3.821 | 25.112 ±6.005 | 23.865 ±5.858 |
| | DFITE | 19.177 ±10.942 | 2.828 ±0.440 | 17.404 ±2.933 | 3.083 ±0.486 | 23.042 ±8.474 | 3.787 ±0.642 |
| | DERCFR | 12.175 ±2.673 | 10.017 ±2.344 | 21.704 ±3.786 | 17.246 ±3.825 | 29.889 ±4.950 | 23.725 ±4.582 |
| | CFR | 27.010 ±11.249 | 2.932 ±0.548 | 24.218 ±3.939 | 2.754 ±0.599 | 32.181 ±14.965 | 3.154 ±0.494 |
| | NetEST | 4.342 ±0.642 | 0.875 ±0.112 | 6.822 ±1.107 | 1.405 ±0.219 | 9.136 ±1.748 | 2.052 ±0.333 |
| | Deconfounder | 6.275 ±0.727 | 3.428 ±0.401 | 8.338 ±1.230 | 4.738 ±0.844 | 10.598 ±1.903 | 6.282 ±0.944 |
| | SPNET | 7.208 ±0.691 | 1.043 ±0.173 | 8.693 ±1.030 | 1.204 ±0.216 | 10.167 ±1.622 | 1.482 ±0.445 |
| | DNDC | 1.892 ±0.142 | 1.314 ±0.154 | 2.589 ±0.959 | 1.618 ±0.781 | 3.154 ±0.998 | 1.881 ±1.090 |
| | **DSPNET** | **1.346 ±0.173**$^{\dagger}$ | **0.788 ±0.111**$^{\dagger}$ | **1.497 ±0.145**$^{\dagger}$ | **0.890 ±0.080**$^{\dagger}$ | **1.805 ±0.288**$^{\dagger}$ | **1.072 ±0.318**$^{\dagger}$ |
| BlogCatalog | DESCN | 13.812 ±2.041 | 12.934 ±2.083 | 23.430 ±3.422 | 22.348 ±3.428 | 32.849 ±6.206 | 31.539 ±6.139 |
| | DFITE | 8.224 ±2.465 | 2.547 ±0.589 | 11.841 ±3.243 | 3.446 ±0.427 | 14.949 ±3.912 | 4.383 ±1.077 |
| | DERCFR | 21.791 ±5.340 | 16.090 ±2.407 | 35.321 ±8.824 | 24.921 ±3.295 | 47.289 ±11.642 | 34.239 ±7.866 |
| | CFR | 8.156 ±2.369 | 0.984 ±0.158 | 11.547 ±3.164 | 1.295 ±0.249 | 14.781 ±3.744 | 1.631 ±0.428 |
| | NetEST | 5.302 ±0.664 | 1.272 ±0.235 | 8.539 ±1.074 | 1.586 ±0.218 | 11.671 ±1.923 | 2.021 ±0.302 |
| | Deconfounder | 7.811 ±1.172 | 5.226 ±0.849 | 13.067 ±1.863 | 8.884 ±1.170 | 18.247 ±3.522 | 12.451 ±2.533 |
| | SPNET | 6.674 ±0.460 | 1.643 ±0.480 | 9.569 ±1.742 | 2.298 ±0.859 | 12.206 ±2.198 | 2.743 ±0.809 |
| | DNDC | 1.961 ±0.127 | 1.314 ±0.155 | 2.475 ±0.462 | 1.454 ±0.400 | 2.905 ±0.290 | 1.597 ±0.144 |
| | **DSPNET** | **1.309 ±0.125**$^{\dagger}$ | **0.853 ±0.093**$^{\dagger}$ | **1.464 ±0.119**$^{\dagger}$ | **0.845 ±0.105**$^{\dagger}$ | **1.773 ±0.426**$^{\dagger}$ | **1.068 ±0.488**$^{\dagger}$ |

## C.3 PERFORMANCE COMPARISON UNDER VARYING INFLUENCE OF CURRENT COVARIATES

As shown in the data generation, $\lambda_2$ is set to control the influence of the current unit's covariates on its hidden confounders $z_i^t$. Here we compare the treatment effect estimation performance between baselines and the proposed model DSPNET under varying values of $\lambda_2$ on Flickr and BlogCatalog. We set the value of $\lambda_2$ to {3,5,8} and remain the other parameter as default.

The experimental results are shown in Table 8. We can see that our proposed model DSPNET outperforms all the baselines under different levels of current covariates' influence. Moreover, it can be observed that under varying levels of influence $\lambda_2$, the proposed model DSPNET exhibits strong stability in estimating treatment effects, as it effectively captures confounders in the latent space and learns the factors that truly affect the outcome.

## C.4 PERFORMANCE COMPARISON UNDER VARYING INFLUENCE OF NETWORK STRUCTURE

In the data generation, $\lambda_3$ is to control the influence of network structure, i.e., unit's neighbors, on its hidden confounders. Similarly, here we conduct the experiments to compare the treatment effect estimation performance between the baselines and the proposed model DSPNET under varying $\lambda_3$ values. We set the value of $\lambda_3$ to also range in {3,5,8} and remain the other parameters in data generation as default.

The experimental results are shown in Table 9. Still, our proposed model DSPNET outperforms the other baselines in all settings of $\lambda_3$ values and the performance of DSPNET remains stable when the influence of network structure varies.

## C.5 ROBUSTNESS STUDY OF UNOBSERVED CONFOUNDERS

We also conduct robustness analyses regarding the unobserved confounders that may not be fully captured. Specifically, we explicitly select 10%, 20%, 30% covariates that affects both treatment and outcome as the latent confounders, and these selected latent confounders are not included in the construction of function $\Phi_z(\cdot)$, thereby simulating scenarios where the confounder modeling module fails to capture all confounders directly.

Table 8: CATE-ID performance comparison by varying influence of current covariates on Flickr and BlogCatalog datasets. **Bold**: the best results. Underline: the 2nd best results. Lower is better. [†] indicates statistically significant improvement over the strongest baseline (t-test, $p$-value $< 0.05$).

| Datasets | Methods | $\lambda_2 = 3$ | | $\lambda_2 = 5$ | | $\lambda_2 = 8$ | |
|---|---|---|---|---|---|---|---|
| | | $\sqrt{\epsilon_{PEHE}}$ | $\epsilon_{ATE}$ | $\sqrt{\epsilon_{PEHE}}$ | $\epsilon_{ATE}$ | $\sqrt{\epsilon_{PEHE}}$ | $\epsilon_{ATE}$ |
| Flickr | DESCN | 17.982 ±3.872 | 16.996 ±3.821 | 14.089 ±2.125 | 13.141 ±2.122 | 11.797 ±2.562 | 10.823 ±2.582 |
| | DFITE | 17.404 ±2.933 | 3.083 ±0.486 | 14.872 ±2.196 | 2.679 ±0.270 | 16.166 ±3.923 | 2.634 ±0.670 |
| | DERCFR | 21.704 ±3.786 | 17.246 ±3.825 | 15.100 ±2.852 | 11.958 ±2.363 | 13.250 ±2.015 | 10.515 ±1.760 |
| | CFR | 24.218 ±3.939 | 2.754 ±0.599 | 23.425 ±2.193 | 3.081 ±0.864 | 23.708 ±2.849 | 3.048 ±0.788 |
| | NetEST | 6.822 ±1.107 | 1.405 ±0.219 | 5.437 ±0.615 | 1.074 ±0.179 | 4.750 ±0.811 | 0.904 ±0.155 |
| | Deconfounder | 8.338 ±1.230 | 4.738 ±0.844 | 7.309 ±0.885 | 4.154 ±0.789 | 6.786 ±1.017 | 3.759 ±0.708 |
| | SPNET | 8.693 ±1.030 | 1.204 ±0.216 | 8.006 ±0.660 | 1.173 ±0.203 | 7.541 ±0.859 | 1.068 ±0.206 |
| | DNDC | 2.589 ±0.959 | 1.618 ±0.781 | 2.338 ±0.533 | 1.561 ±0.449 | 2.100 ±0.361 | 1.361 ±0.281 |
| | **DSPNET** | **1.497 ±0.145**[†] | **0.890 ±0.080**[†] | **1.440 ±0.137**[†] | **0.825 ±0.129**[†] | **1.373 ±0.151**[†] | **0.784 ±0.140**[†] |
| BlogCatalog | DESCN | 23.430 ±3.422 | 22.348 ±3.428 | 19.826 ±5.019 | 18.804 ±4.882 | 18.436 ±4.117 | 17.393 ±3.978 |
| | DFITE | 11.841 ±3.243 | 3.446 ±0.427 | 11.050 ±4.279 | 3.222 ±0.471 | 12.446 ±4.225 | 2.866 ±0.437 |
| | DERCFR | 35.321 ±8.824 | 24.921 ±3.295 | 29.706 ±6.265 | 22.365 ±5.321 | 31.197 ±6.479 | 21.824 ±4.618 |
| | CFR | 11.547 ±3.164 | 1.295 ±0.249 | 10.574 ±4.037 | 1.033 ±0.207 | 12.396 ±4.525 | 1.021 ±0.144 |
| | NetEST | 8.539 ±1.074 | 1.586 ±0.218 | 7.205 ±1.390 | 1.422 ±0.188 | 6.675 ±1.177 | 1.332 ±0.176 |
| | Deconfounder | 13.067 ±1.863 | 8.884 ±1.170 | 11.527 ±3.180 | 7.659 ±2.202 | 10.195 ±3.013 | 6.600 ±1.901 |
| | SPNET | 9.569 ±1.742 | 2.298 ±0.859 | 8.290 ±1.515 | 1.733 ±0.413 | 8.421 ±1.420 | 2.235 ±0.551 |
| | DNDC | 2.475 ±0.462 | 1.454 ±0.400 | 2.180 ±0.182 | 1.260 ±0.189 | 2.273 ±0.348 | 1.517 ±0.416 |
| | **DSPNET** | **1.464 ±0.119**[†] | **0.845 ±0.105**[†] | **1.549 ±0.044**[†] | **0.852 ±0.083**[†] | **1.573 ±0.129**[†] | **0.951 ±0.147**[†] |

Table 9: CATE-ID performance comparison by varying influence of network structure on Flickr and BlogCatalog datasets. **Bold**: the best results. Underline: the 2nd best results. Lower is better. [†] indicates statistically significant improvement over the strongest baseline (t-test, $p$-value $< 0.05$).

| Datasets | Methods | $\lambda_3 = 3$ | | $\lambda_3 = 5$ | | $\lambda_3 = 8$ | |
|---|---|---|---|---|---|---|---|
| | | $\sqrt{\epsilon_{PEHE}}$ | $\epsilon_{ATE}$ | $\sqrt{\epsilon_{PEHE}}$ | $\epsilon_{ATE}$ | $\sqrt{\epsilon_{PEHE}}$ | $\epsilon_{ATE}$ |
| Flickr | DESCN | 17.982 ±3.872 | 16.996 ±3.821 | 16.726 ±2.266 | 15.802 ±2.203 | 10.468 ±1.807 | 9.648 ±1.806 |
| | DFITE | 17.404 ±2.933 | 3.083 ±0.486 | 20.351 ±8.152 | 3.076 ±0.611 | 16.333 ±5.052 | 2.649 ±0.393 |
| | DERCFR | 21.704 ±3.786 | 17.246 ±3.825 | 17.721 ±1.587 | 14.079 ±1.777 | 12.066 ±2.430 | 9.898 ±2.171 |
| | CFR | 24.218 ±3.939 | 2.754 ±0.599 | 28.309 ±7.360 | 3.106 ±0.649 | 27.616 ±9.904 | 3.171 ±0.524 |
| | NetEST | 6.822 ±1.107 | 1.405 ±0.219 | 6.222 ±0.662 | 1.256 ±0.217 | 4.278 ±0.501 | 0.842 ±0.144 |
| | Deconfounder | 8.338 ±1.230 | 4.738 ±0.844 | 8.082 ±0.770 | 4.607 ±0.601 | 6.273 ±0.665 | 3.428 ±0.550 |
| | SPNET | 8.693 ±1.030 | 1.204 ±0.216 | 8.432 ±0.549 | 1.238 ±0.251 | 7.031 ±0.626 | 0.982 ±0.200 |
| | DNDC | 2.589 ±0.959 | 1.618 ±0.781 | 2.273 ±0.217 | 1.485 ±0.176 | 1.946 ±0.317 | 1.322 ±0.244 |
| | **DSPNET** | **1.497 ±0.145**[†] | **0.890 ±0.080**[†] | **1.565 ±0.171**[†] | **0.939 ±0.137**[†] | **1.305 ±0.088**[†] | **0.770 ±0.092**[†] |
| BlogCatalog | DESCN | 23.430 ±3.422 | 22.348 ±3.428 | 18.261 ±3.824 | 17.300 ±3.641 | 12.651 ±2.103 | 11.815 ±2.011 |
| | DFITE | 11.841 ±3.243 | 3.446 ±0.427 | 10.308 ±4.692 | 2.867 ±0.600 | 7.579 ±1.930 | 2.311 ±0.580 |
| | DERCFR | 35.321 ±8.824 | 24.921 ±3.295 | 27.508 ±7.344 | 20.028 ±4.403 | 18.786 ±4.341 | 13.430 ±2.645 |
| | CFR | 11.547 ±3.164 | 1.295 ±0.249 | 10.060 ±5.000 | 1.079 ±0.225 | 7.198 ±2.051 | 0.865 ±0.159 |
| | NetEST | 8.539 ±1.074 | 1.586 ±0.218 | 6.648 ±1.136 | 1.563 ±0.132 | 4.815 ±0.683 | 1.219 ±0.196 |
| | Deconfounder | 13.067 ±1.863 | 8.884 ±1.170 | 10.264 ±1.967 | 6.707 ±1.373 | 7.156 ±1.385 | 4.605 ±0.987 |
| | SPNET | 9.569 ±1.742 | 2.298 ±0.859 | 7.918 ±1.529 | 1.796 ±0.475 | 6.575 ±1.327 | 1.544 ±0.441 |
| | DNDC | 2.475 ±0.462 | 1.454 ±0.400 | 2.057 ±0.137 | 1.250 ±0.129 | 1.800 ±0.214 | 1.078 ±0.109 |
| | **DSPNET** | **1.464 ±0.119**[†] | **0.845 ±0.105**[†] | **1.521 ±0.129**[†] | **0.870 ±0.110**[†] | **1.316 ±0.096**[†] | **0.790 ±0.102**[†] |

Table 10: DSPNET performance under different unobserved confounding levels.

| Level | Flickr | | BlogCatalog | |
|---|---|---|---|---|
| | $\sqrt{\epsilon_{PEHE}}$ | $\epsilon_{ATE}$ | $\sqrt{\epsilon_{PEHE}}$ | $\epsilon_{ATE}$ |
| 10% | $1.535 \pm 0.133$ | $0.911 \pm 0.102$ | $1.613 \pm 0.188$ | $0.949 \pm 0.206$ |
| 20% | $1.694 \pm 0.102$ | $0.968 \pm 0.081$ | $1.697 \pm 0.089$ | $1.009 \pm 0.064$ |
| 30% | $1.712 \pm 0.129$ | $1.112 \pm 0.093$ | $1.830 \pm 0.194$ | $1.021 \pm 0.212$ |
| original (0%) | $\mathbf{1.497 \pm 0.145}$ | $\mathbf{0.890 \pm 0.080}$ | $\mathbf{1.464 \pm 0.119}$ | $\mathbf{0.845 \pm 0.105}$ |

Table 11: Performance of DSPNET and network-based baselines under different unobserved confounding levels.

| Dataset | Model | Level = 10% | | Level = 20% | | Level = 30% | |
|---|---|---|---|---|---|---|---|
| | | $\sqrt{\epsilon_{PEHE}}$ | $\epsilon_{ATE}$ | $\sqrt{\epsilon_{PEHE}}$ | $\epsilon_{ATE}$ | $\sqrt{\epsilon_{PEHE}}$ | $\epsilon_{ATE}$ |
| Flickr | Deconfounder | $8.350 \pm 1.324$ | $4.807 \pm 0.942$ | $8.578 \pm 1.375$ | $4.950 \pm 1.003$ | $8.810 \pm 0.998$ | $5.216 \pm 0.667$ |
| | NetEST | $6.982 \pm 1.121$ | $1.484 \pm 0.204$ | $7.172 \pm 1.259$ | $1.590 \pm 0.189$ | $7.538 \pm 1.375$ | $1.610 \pm 0.219$ |
| | SPNET | $8.870 \pm 1.123$ | $1.334 \pm 0.256$ | $9.045 \pm 1.260$ | $1.355 \pm 0.304$ | $9.233 \pm 1.453$ | $1.398 \pm 0.210$ |
| | DNDC | $2.637 \pm 0.8878$ | $1.724 \pm 0.581$ | $2.749 \pm 0.752$ | $1.751 \pm 0.585$ | $2.878 \pm 0.604$ | $1.768 \pm 0.592$ |
| | DSPNET | $\mathbf{1.535 \pm 0.133}$ | $\mathbf{0.911 \pm 0.102}$ | $\mathbf{1.694 \pm 0.102}$ | $\mathbf{0.968 \pm 0.081}$ | $\mathbf{1.712 \pm 0.129}$ | $\mathbf{1.112 \pm 0.093}$ |
| BlogCatalog | Deconfounder | $13.231 \pm 1.839$ | $8.932 \pm 1.212$ | $13.773 \pm 1.792$ | $8.978 \pm 1.280$ | $13.906 \pm 2.092$ | $9.126 \pm 1.483$ |
| | NetEST | $8.659 \pm 1.065$ | $1.632 \pm 0.319$ | $8.859 \pm 1.095$ | $1.752 \pm 0.319$ | $9.132 \pm 1.146$ | $1.827 \pm 0.322$ |
| | SPNET | $9.725 \pm 1.546$ | $2.227 \pm 0.838$ | $9.882 \pm 1.346$ | $2.681 \pm 0.846$ | $10.379 \pm 1.475$ | $2.782 \pm 0.943$ |
| | DNDC | $2.571 \pm 0.336$ | $1.527 \pm 0.486$ | $2.748 \pm 0.364$ | $1.684 \pm 0.388$ | $2.906 \pm 0.440$ | $1.692 \pm 0.407$ |
| | DSPNET | $\mathbf{1.613 \pm 0.188}$ | $\mathbf{0.949 \pm 0.206}$ | $\mathbf{1.697 \pm 0.089}$ | $\mathbf{1.009 \pm 0.064}$ | $\mathbf{1.830 \pm 0.194}$ | $\mathbf{1.021 \pm 0.212}$ |

The CATE-ID estimation performance of DSPNET for different levels of unobserved confounding is shown in Table 10. Besides, we also compare the CATE-ID estimation of DSPNET with those network-based baselines (NetEST, Deconfounder, SPNET and DNDC) under different levels of unobserved confounding, as shown in Table 11. We can see that DSPNET remains stable and continues to outperform baselines under different unobserved confounding levels. The above results indicate that our method is robust even when there are unobserved confounders, as the proposed model can infer the dynamic latent confounders by leveraging the network structure and neighbors' characteristics.

## C.6 EMPIRICAL STUDY OF MODEL COMPLEXITY

We conduct an empirical study to examine how increasing the architectural depth and width affects the performance of DSPNET. Specifically, we independently explored three key architectural hyperparameters about model complexity: (1) the depth of the GCN backbone, (2) the depth of the MLP backbone, (3) the size of the hidden dimensions.

For each hyperparameter, we varied one factor while fixing the remaining two to their default settings. Concretely, we evaluated GCN depths of 1, 2, 3 layers, MLP depths of 3, 4, 5 layers, and hidden dimensions of 100, 150, 200. The performance of DSPNET for CATE-ID estimation on Flickr and BlogCatalog in different architectural settings is shown in Table 12.

The results show that the performance of DSPNET under different MLP layers and hidden dimensions remains stable across all tested configurations. However, adding more GCN layers leads to a clear performance drop due to the well-known over-smoothing effect, where deeper GCNs produce increasingly similar node representations and erase individual-level heterogeneity essential for accurate CATE estimation. Moreover, in many networks, 1-hop neighbors already provide sufficient information, while deeper propagation introduces irrelevant signals from distant nodes, further degrading performance.

## C.7 PARAMETER ANALYSIS OF $\omega$ CONTROLLING OVER-FITTING TERM

In the hyperparameter analysis of the main body, we only investigate the impact of the hyperparameter $\alpha$ (control the influence of treatment prediction) and $\beta$ (control the contribution of gradient reversal

Table 12: Empirical study of model complexity.

| Setting | Value | Flickr | | BlogCatalog | |
|---|---|---|---|---|---|
| | | $\sqrt{\epsilon_{PEHE}}$ | $\epsilon_{ATE}$ | $\sqrt{\epsilon_{PEHE}}$ | $\epsilon_{ATE}$ |
| # of GCN Layers | 1 | 1.497±0.145 | 0.890±0.080 | 1.464±0.119 | 0.845±0.105 |
| | 2 | 5.299±4.407 | 2.543±2.811 | 6.293±4.441 | 3.091±3.803 |
| | 3 | 15.703±3.403 | 6.714±2.826 | 16.868±7.206 | 8.758±7.103 |
| # of MLP Layers | 3 | 1.455±0.145 | 0.814±0.110 | 1.355±0.145 | 0.814±0.110 |
| | 4 | 1.571±0.107 | 0.918±0.116 | 1.371±0.107 | 0.858±0.116 |
| | 5 | 1.416±0.149 | 0.850±0.116 | 1.486±0.149 | 0.850±0.116 |
| # of Hidden Dim | 100 | 1.435±0.121 | 0.887±0.087 | 1.412±0.110 | 0.812±0.141 |
| | 150 | 1.373±0.103 | 0.836±0.089 | 1.419±0.221 | 0.772±0.130 |
| | 200 | 1.436±0.159 | 0.844±0.084 | 1.577±0.433 | 0.827±0.154 |

component), here we further conduct additional experiments on analyzing the impact of $\omega$ controlling the over-fitting term in Eq.(10) on the proposed model DSPNET.

We set $\omega$ to range in $\{10^{-4}, 10^{-3}, 10^{-2}, 10^{-1}, \}$ and plot the curve chart of CATE-ID estimation performance in terms of $\sqrt{\epsilon_{PEHE}}$ and $\epsilon_{ATE}$ on Flickr and BlogCatalog. The results are shown in Figure 5, we can see that the proposed DSPNET performs steadily under varying values of $\omega$ in both $\sqrt{\epsilon_{PEHE}}$ and $\epsilon_{ATE}$, which means that DSPNET is not sensitive to the hyperparameter $\omega$ controlling over-fitting term.

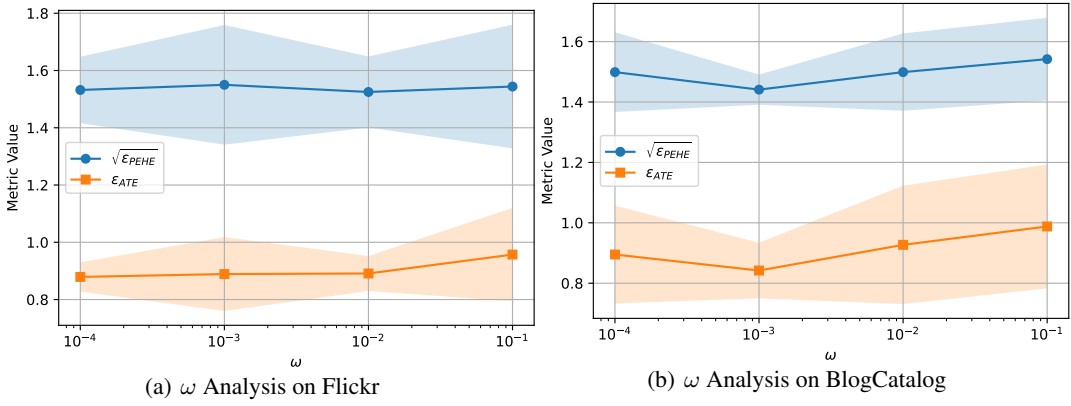

(a) $\omega$ Analysis on Flickr

(b) $\omega$ Analysis on BlogCatalog

Figure 5: Hyperparameter Analysis of $\omega$.

## C.8 COMPARISON FOR DIFFERENT BALANCING STRATEGIES

Apart from the gradient reversal method adopted in DSPNET, another common approach to mitigate confounding bias is to enforce distributional balance of confounder representations in the latent space. Two widely used balancing strategies include the Wasserstein-1 (Wass) distance and Maximum Mean Discrepancy (MMD). Both aim to minimize the distributional divergence between the treated and control groups with respect to the confounder representations, thereby ensuring that the learned confounders are not predictive of the treatment assignment.

To demonstrate the effectiveness of the gradient reversal strategy used in our model, we compare it against two widely used balancing strategies—Wasserstein-1 (Wass) distance and Maximum Mean Discrepancy (MMD)—in the task of treatment effect estimation on Flickr and BlogCatalog. As shown in Figure 6, our gradient reversal consistently outperforms the alternatives in different metrics and datasets.

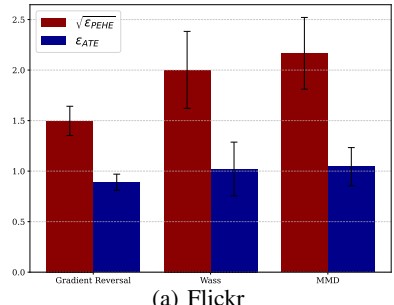
(a) Flickr

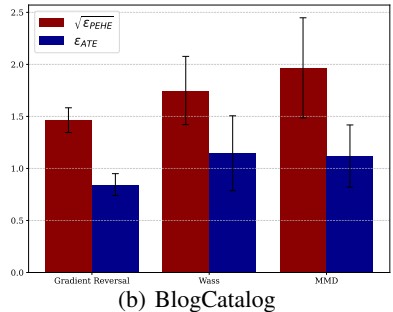
(b) BlogCatalog

Figure 6: Treatment effect estimation performance of DSPNET under different balance strategies.

Table 13: Performance comparison between unit GRU and LSTM.

| Datasets | Model | $p\% = 0.1\%$ | | $p\% = 0.5\%$ | | $p\% = 1.0\%$ | |
|---|---|---|---|---|---|---|---|
| | | $\sqrt{\epsilon_{PEHE}}$ | $\epsilon_{ATE}$ | $\sqrt{\epsilon_{PEHE}}$ | $\epsilon_{ATE}$ | $\sqrt{\epsilon_{PEHE}}$ | $\epsilon_{ATE}$ |
| Flickr | DSPNET$_{GRU}$ | 1.497$\pm$0.145 | 0.890$\pm$0.080 | 2.062$\pm$0.498 | 1.144$\pm$0.128 | 2.189$\pm$0.205 | 1.351$\pm$0.209 |
| | DSPNET$_{LSTM}$ | 1.498$\pm$0.119 | 0.871$\pm$0.099 | 2.181$\pm$0.419 | 1.227$\pm$0.332 | 2.023$\pm$0.137 | 1.276$\pm$0.150 |
| BlogCatalog | DSPNET$_{GRU}$ | 1.464$\pm$0.119 | 0.845$\pm$0.105 | 1.506$\pm$0.237 | 0.913$\pm$0.204 | 2.227$\pm$0.378 | 1.183$\pm$0.290 |
| | DSPNET$_{LSTM}$ | 1.517$\pm$0.171 | 0.906$\pm$0.195 | 1.512$\pm$0.113 | 0.938$\pm$0.176 | 2.260$\pm$0.502 | 0.948$\pm$0.145 |

## C.9 COMPARISON OF GRU AND LSTM UNITS FOR ENCODING HISTORICAL INFORMATION

In the experiments in main body, the proposed DSPNET employs a Gated Recurrent Unit (GRU) (Cho et al., 2014) to encode historical information by capturing hidden confounders, treatment assignments, and past temporal states. While GRUs are computationally efficient and effective for modeling sequential dependencies, another widely used recurrent architecture is the Long Short-Term Memory (LSTM) network (Hochreiter & Schmidhuber, 1997). To assess whether the performance of DSPNET depends on the choice of recurrent unit, we conducted an ablation study comparing GRU and LSTM as alternative modules for encoding historical information.

We replaced the GRU cell in DSPNET with an LSTM cell while keeping all other components, hyperparameters, and training configurations unchanged. Here we report the comparison results between GRU and LSTM cell on Flickr and BlogCatalog by default generation for varying degress of network dynamics. As shown in Table 13, one can see that both GRU- and LSTM-based variants of DSPNET achieve comparable performance.

## C.10 DIFFERENT EXPOSURE FUNCTIONS

In real-world networks, interference depends not only on neighbors' treatment assignments, but also on how these treatments interact with intrinsic behavioral patterns of individuals. For example, a health intervention (e.g., jog) may affect a user only if its treated neighbors actively engage in and share health-related behaviors. Hence, simple exposure summaries (e.g., averaging neighbor treatments) oversimplify these dynamics. Unlike prior work, DSPNET leverages treatment assignments as gating signals and builds a learnable interference representation through an additional GCN module, which serves as the exposure summary function for modeling interference, as shown in Equation (6).

To examine how sensitive DSPNET is to the design of the summary function, we conducted experiments where we varied the aggregation rule of DSPNET, we derive the following three variants of exposure summary function to capture the interference from neighbors:

- **Sum-Pooling**: directly sum of the treatment values of unit $i$'s neighbors, thus the exposure summary of unit $i$ is formulated as $e_i^t = \sum_{j \in \mathcal{G}_i^t} d_j^t$.

- **Average-Pooling**: average the treatment assignments of unit $i$'s neighbors, then the exposure summary of unit $i$ is formulated as $e_i^t = \frac{\sum_{j \in \mathcal{G}_i^t} d_j^t}{|\mathcal{G}_i^t|}$.

Table 14: Comparison results for different variants of exposure summary function.

| Variants | Flickr | | BlogCatalog | |
|---|---|---|---|---|
| | $\sqrt{\epsilon_{PEHE}}$ | $\epsilon_{ATE}$ | $\sqrt{\epsilon_{PEHE}}$ | $\epsilon_{ATE}$ |
| *Sum-Pooling* | 1.570 ±0.190 | 1.163 ±0.162 | 1.752 ±0.166 | 1.183 ±0.182 |
| *Average-Pooling* | 1.695 ±0.152 | 1.142 ±0.135 | 1.543 ±0.164 | 1.073 ±0.198 |
| *PS-Weighting* | 1.598 ±0.121 | 1.121 ±0.135 | 1.502 ±0.062 | 0.891 ±0.091 |
| *Original* | **1.497± 0.145** | **0.890± 0.080** | **1.464 ±0.119** | **0.845 ±0.105** |

- **PS-Weighting**: use each unit's neighbors' propensity scores as weights to aggregate the GCN-based neighbor representations as the summary exposure: $\boldsymbol{e}_i^t = \sum_{j \in \mathcal{G}_i^t} \pi(\boldsymbol{x}_j^t) \cdot \boldsymbol{r}_j^t$, where $\pi(\boldsymbol{x}_j^t)$ is the propensity score, here we use the MLP network with softmax function to predict the propensity scores.

Then we compare the variants of the above three summary functions with the original strategy of DSPNET on Flickr and BlogCatalog with default generation for the CATE-ID estimation task. The comparison results are shown in Table 14. We can see that the exposure function with representations (i.e., the PS-Weighting and Original) is relatively better, indicating that considering the heterogeneity of neighbor's interference is important, but the other variants are also competitive.

### C.11 CONFOUNDER DISTRIBUTION VISUALIZATION

Additionally, we compared the distributions of confounder representations learned by the DSPNET model, both with and without the balancing strategy, i.e., the gradient reversal component. In particular, we randomly sampled representations of control and treated group samples at a selected time step from models trained under conditions with and without the balancing strategy. These representations were then reduced to two dimensions using t-SNE, and visualized through scatter plots. As illustrated in the Figure 7, the confounder representations of the treated and control groups obtained using the balancing strategy are more closely clustered compared to those without the balancing strategy, indicating a smaller distance between the two groups.

## D LIMITATION

In this work, our proposed approach explicitly models local interference, assuming that only treatments assigned to immediate neighbors influence an individual's outcome. While this simplifies analysis and computational complexity, it inherently neglects global interference effects—scenarios where units beyond direct neighbors can impact the outcomes through indirect or cascading pathways within the network. Such global spillover effects are plausible in many real-world settings, particularly in dense or highly interconnected networks, and their omission could potentially lead to incomplete or biased causal estimates. Therefore, extending our approach to account for broader network influence remains an important direction for future research.

And our work assumes that a given unit's treatment status does not influence the treatment assignments of its neighbors, we only consider the interference which refers to the influence of one unit's neighbors' treatment on its outcome. However, in real-world networked settings, treatments may spread contagiously or through social influence processes, violating this assumption. Ignoring such treatment dependence could limit the generalizability of our findings. Incorporating treatment contagion mechanisms into the modeling framework represents another valuable avenue for future research.

Furthermore, in observational social networks, covariates and structure are often entangled through homophily. And homophily and contagion (peer influence, interference) are fundamentally confounded in observational settings, making it impossible to cleanly distinguish "similarity due to shared attributes" from "similarity due to peer influence". In this work, we does not aim to identify or separate homophily from contagion, but we acknowledge that fully separating homophily-driven sim-

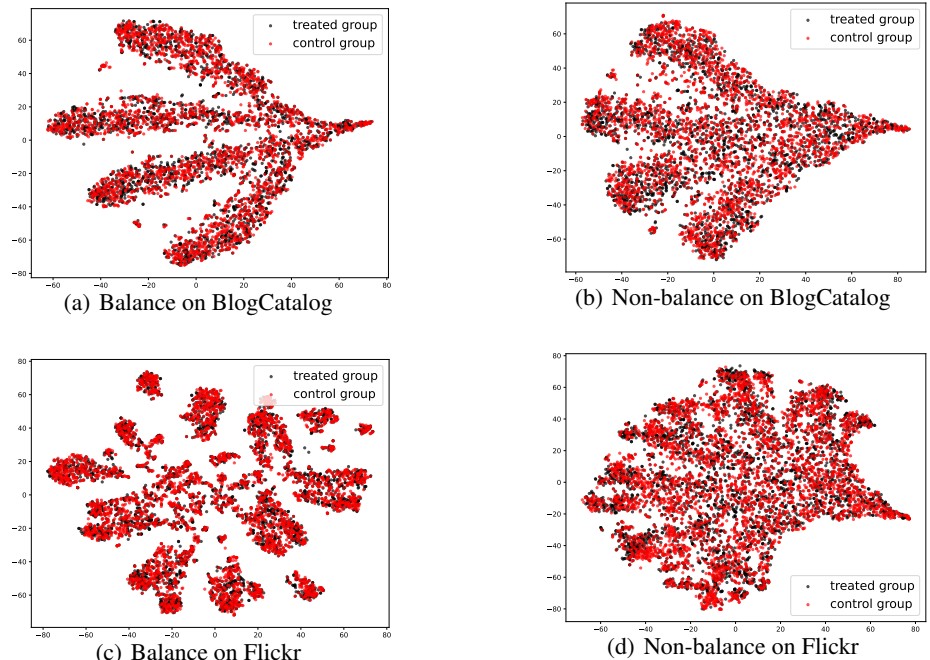

Figure 7: Confounder distribution visualization with and without balancing strategy on BlogCatalog and Flickr.

ilarity from contagion-driven influence in dynamic observational networks is an open and important problem.

# E   LLM USAGE

We used large language models (LLMs) solely to aid in the writing process, including polishing grammar and improving clarity of exposition. No part of the research design, theoretical development, experiments, or analysis relied on LLMs.

