# OpenReview forum: "Modeling Interference for Treatment Effect Estimation in Network Dynamic Environment"
_ICLR.cc/2026/Conference — ICLR 2026 Poster_

### Official Review · Reviewer_mUQv · 2025-10-30

**Soundness:** 3
**Presentation:** 3
**Contribution:** 3
**Rating:** 6
**Confidence:** 2

**Summary:**

Overall, this paper offers a **technically sound and empirically thorough** approach to a timely and underexplored problem — causal inference in dynamic network environments with interference.
Key strengths include the introduction of a formally identifiable estimand (Eq. 2; Sec. 3), a well-structured architecture combining temporal and graph components (Fig. 2; Sec. 4), and strong empirical results (Table 1, Fig. 3).
However, the **proof of identifiability** relies on strong and untested assumptions (Assumptions 3.1–3.2), and the **causal interpretation** of learned embeddings (especially the interference term ( e_t^i )) remains opaque.
The **datasets are synthetic**, which limits the generalizability of findings.
Overall, the work is **technically competent and innovative**, but it needs clearer causal justification and stronger empirical grounding.

**Strengths:**

* **Novel causal estimand and theory** — The paper proposes CATE-ID (Eq. 2) to extend CATE to dynamic settings with interference and proves its identifiability under explicit assumptions (Theorem 3.3). This formalization fills a theoretical gap in existing literature (Sec. 3).
* **Architectural integration of temporal and relational modeling** — DSPNET elegantly combines GCNs (Eq. 4) and GRUs (Eq. 5) to model time-varying confounders, capturing historical and neighborhood dependencies.
* **Explicit interference modeling** — The environment exposure variable ( e_t^i = \sum_{j \in G_i^t} d_t^j r_t^j ) (Eq. 6) effectively parameterizes spillover effects rather than using static aggregation (Sec. 4.2).
* **Adversarial confounder balancing** — The GRL mechanism (Eqs. 9–11) applies domain-adversarial learning to minimize bias across treatment groups, grounded in causal representation learning theory (Shalit et al., 2017).

**Weaknesses:**

* **Overly strong identifiability assumptions** — The Extended Ignorability (Assumption 3.1) and Consistency (Assumption 3.2) presuppose that all confounders are captured by Φ_z(·) learned via GCNs. No empirical verification or sensitivity analysis is given.
* **Causal interpretation is weak** — The interference representation ( e_t^i ) is learned implicitly but lacks interpretability or causal diagnostics (e.g., no analysis of which neighbor interactions drive outcomes).
* **Limited realism in evaluation** — All experiments are synthetic with simulated treatments, outcomes, and interference (Appendix A). No semi-synthetic or real-world dataset (e.g., citation, epidemiological, or economic network) is used.
* **Scalability untested** — There is no complexity or runtime comparison (Sec. 5.3 omits computational benchmarks), and DSPNET’s feasibility on large-scale graphs is unclear.
* **No uncertainty estimation** — The model reports point estimates without confidence intervals or statistical tests, which is critical in causal inference.
* **Potential redundancy with prior works** — DSPNET’s components (GCN+GRU+GRL) resemble DNDC (Ma et al., 2021) and SPNET (Huang et al., 2023); the novelty mainly lies in combining them rather than introducing fundamentally new mechanisms.

**Questions:**

1. **Validity of the Identifiability Assumptions**
   The identifiability proof of CATE-ID (Sec. 3.2, Theorem 3.3) hinges on the Extended Ignorability and Consistency assumptions (Assumptions 3.1–3.2).
   Could you clarify **how these assumptions are justified or empirically supported** in your experimental setup?
   For example, are all relevant confounders explicitly simulated and observed in the synthetic data, and how sensitive is DSPNET to violations of these assumptions?

2. **Interpretability of the Interference Representation ( e_t^i )**
   The model learns a dynamic exposure representation ( e_t^i ) through neighborhood aggregation (Eq. 6).
   Could you provide **qualitative or quantitative analyses** showing what this embedding captures?
   For instance, does it correlate with known structural properties (e.g., degree, clustering, influence centrality), or can it be interpreted as a measurable causal quantity such as “average neighbor treatment intensity”?

3. **Comparison with Prior Dynamic Network Models**
   DSPNET combines GCN, GRU, and adversarial balancing layers.
   Could you **clarify the novelty over DNDC (Ma et al., 2021)** and **SPNET (Huang et al., 2023)** beyond architectural integration?
   A detailed ablation or theoretical justification isolating what enables DSPNET to outperform these baselines would help clarify its unique contribution.

4. **Realism and Generalizability of the Evaluation**
   All experiments are conducted on synthetic data (Flickr and BlogCatalog dynamic variants).
   Could you comment on **how realistic these synthetic dynamics are**, and whether DSPNET could be applied to real-world dynamic networks (e.g., temporal citation graphs, contact tracing, or recommendation systems)?
   Are there any known limitations or scalability barriers when moving from synthetic to real data?

---

> ### Author Response · Authors · 2025-11-21
> **The response to Reviewer mUQv: [W1 And Q1]**
>
> We sincerely appreciate the reviewer’s great efforts and thoughtful comments, which greatly help improve our manuscript. Below, we address the concerns and questions point by point and try our best to update the manuscript accordingly, these updates are temporarily highlighted in $\textcolor{blue}{blue}$ to facilitate the reviewer’s checking.
>
> > **W1 and Q1: Regarding the Validity of Identifiability Assumptions (Consistency and Extended Ignorability Assumption).**
>
> **Response**: Thank you for your comments. We would like to clarify the justification of Assumptions 3.1–3.2 and provide new empirical verification results  as follows:
>
> **(1) Justification of  Consistency Assumption**
>
> - **Standard in potential-outcome framework**. The Consistency assumption is **a foundational assumption** in Neyman–Rubin causal framework and widely adopted in causal study.
> - **Required for identifiability of CATE-ID under interference**.  Consistency Assumption ensures the connection between the observed outcomes and the potential outcomes under the realized treatment–exposure pair, which is essential for identifying the CATE-ID under interference.
>
> **(2) Justification of  Extended Ignorability Assumption**
>
> - First, following previous causal literature in network setting (e.g., Guo et al.; Ma et al., 2021; Huang et al., 2023), we assume that although latent confounders cannot be directly observed, **they can be inferred from observable covariates, network structure, and neighbor characteristics**:
>
>   - For instance, socioeconomic status (SES) is a typical latent confounder that cannot be directly measured but can be inferred from the occupations, education levels, or income patterns of one’s social contacts.
>
>   - Motivated by this, we posit the existence of a representation function $\Phi_z(\cdot)$ that learns a latent variable $Z_i^t$ capturing these underlying confounding factors.
>
> - Then,  this assumption is conceptually aligned with the **standard ignorability assumption** which assumes that all confounders are measured in the observed covariates, our extended ignorability assumes that all confounders are absorbed into the learned latent factor $Z_i^t$, which is **a natural extension of the standard ignorability assumption** and is also implicitly adopted in prior works (e.g., Veitch et al., 2019; Guo et al., 2020;  Ma et al., 2021; Huang et al., 2023) on confounder representation learning.
>
> - Crucially, the Extended Ignorability Assumption is ** required for establishing the theoretical identifiability of CATE-ID**, just as the standard **standard ignorability assumption** is a necessary condition for CATE identifiability in non-network settings (Hill, J. 2011, Shalit et al. 2017, Yao et al. 2018 ).
>
> **(3) Justification in Data Generation**
>
> Our synthetic data generator follows prior work (Ma et al., 2021)  and is designed such that these assumptions hold in the generative process:
> - The generator integrates the observed covariates and network structure to generate the confounders by the latent mapping function, ensuring that confounders are reflected in observed node covariates and the characteristics of neighbors, making them **recoverable through  a latent function $\Phi_z(\cdot)$**.
>
>  - And we utilize the q-order autoregressive process (Mills, 1990; Melnychuk et al, 2022;  Bica et al, 2020) to ensure the **temporal dependence** assumed under Extended Ignorability assumption. And we also utilize the latent mapping function and neighbors' treatments to simulate the dynamic interference (see Eq.(17) Appendix A).
>
> - Then the generator directly uses **all the generated confounders** and **interference** to simulate the treatment and outcome by two structural equations (see Eq.(16) and Eq.(18) in Appendix A). Thus, the generation mechanism is explicitly aligned with the assumptions we adopted in this work.

---

> > ### Author Response · Authors · 2025-11-21
> > **Continue for [W1 And Q1]**
> >
> > **(4) Empirical investigation under Violation of Extended Ignorability Assumption**
> >
> > To address your concern about the potential violation of Extended Ignorability Assumption, we conduct a sensitivity analysis when there exist unobserved confounders that may not be fully captured.
> >
> > Specifically, we explicitly select {10%, 20%, 30%} covariates  that affects both treatment and outcome as the latent confounders, and these selected latent confounders are **not included** in the construction of  function $\Phi_z(\cdot)$,  thereby simulating scenarios where the confounder modeling module fails to capture all confounders directly.
> >
> > - First, the CATE-ID estimation performance of  DSPNET for different levels of unobserved confounding is as follows:
> >
> > |                            |           Flickr           |                        |        BlogCatalog        |                      |
> > | :-------------------------: | :------------------------: | :---------------------: | :------------------------: | :-------------------: |
> > | **Level** | $\sqrt{\epsilon_{PEHE}}$ |   $\epsilon_{ATE}$   | $\sqrt{\epsilon_{PEHE}}$ |  $\epsilon_{ATE}$  |
> > |             10%             |      1.535 $\pm$0.133      |    0.911 $\pm$0.102    |      1.613 $\pm$0.188      |   0.949 $\pm$0.206   |
> > |             20%             |      1.694 $\pm$0.102      |    0.968 $\pm$0.081    |      1.697 $\pm$0.089      |   1.009 $\pm$0.064   |
> > |             30%             |      1.712 $\pm$0.129      |    1.112 $\pm$0.093    |      1.830 $\pm$0.194      |   1.021 $\pm$0.212   |
> > |        original(0%)        |  1.497$\pm$ 0.145  | 0.890$\pm$ 0.080 |   1.464 $\pm$0.119   | 0.845 $\pm$0.105 |
> >
> > - Besides, we also compare the CATE-ID estimation of DSPNET with those network-based baselines (NetEST, Deconfounder, and DNDC) under unobserved confounding:
> >
> > |   Dataset   |     Model    |       Level = 10\%       |                   |       Level = 20\%       |                   |       Level = 30\%       |                   |
> > |:-----------:|:------------:|:------------------------:|:-----------------:|:------------------------:|:-----------------:|:------------------------:|:-----------------:|
> > |   Dataset   |     Model    | $\sqrt{\epsilon_{PEHE}}$ |  $\epsilon_{ATE}$ | $\sqrt{\epsilon_{PEHE}}$ |  $\epsilon_{ATE}$ | $\sqrt{\epsilon_{PEHE}}$ |  $\epsilon_{ATE}$ |
> > |    Flickr   | Deconfounder |     8.350 $\pm$ 1.324    | 4.807 $\pm$ 0.942 |     8.578 $\pm$ 1.375    | 4.950 $\pm$ 1.003 |     8.810 $\pm$ 0.998    | 5.216 $\pm$ 0.667 |
> > |             |    NetEST    |      6.982$\pm$1.121     |  1.484$\pm$0.204  |      7.172$\pm$1.259     |  1.590$\pm$0.189  |      7.538$\pm$1.375     |  1.610$\pm$0.219  |
> > |             |     SPNET    |      8.870$\pm$1.123     |  1.334$\pm$0.256  |      9.045$\pm$1.260     |  1.355$\pm$0.304  |      9.233$\pm$1.453     |  1.398$\pm$0.210  |
> > |             |     DNDC     |     2.637$\pm$0.8878     |  1.724$\pm$0.581  |      2.749$\pm$0.752     |  1.751$\pm$0.585  |      2.878$\pm$0.604     |  1.768$\pm$0.592  |
> > |             |    DSPNET    |      **1.535$\pm$0.133**     |  **0.911$\pm$0.102**  |      **1.694$\pm$0.102**     |  **0.968$\pm$0.081**  |      **1.712$\pm$0.129**     |  **1.112$\pm$0.093**  |
> > | BlogCatalog | Deconfounder |    13.231 $\pm$ 1.839    | 8.932 $\pm$ 1.212 |    13.773 $\pm$ 1.792    | 8.978 $\pm$ 1.280 |    13.906 $\pm$ 2.092    | 9.126 $\pm$ 1.483 |
> > |             |    NetEST    |      8.659$\pm$1.065     |  1.632$\pm$0.319  |      8.859$\pm$1.095     |  1.752$\pm$0.319  |      9.132$\pm$1.146     |  1.827$\pm$0.322  |
> > |             |     SPNET    |      9.725$\pm$1.546     |  2.227$\pm$0.838  |      9.882$\pm$1.346     |  2.681$\pm$0.846  |     10.379$\pm$1.475     |  2.782$\pm$0.943  |
> > |             |     DNDC     |      2.571$\pm$0.336     |  1.527$\pm$0.486  |      2.748$\pm$0.364     |  1.684$\pm$0.388  |      2.906$\pm$0.440     |  1.692$\pm$0.407  |
> > |             |    DSPNET    |      **1.613$\pm$0.188**     |  **0.949$\pm$0.206**  |      **1.697$\pm$0.089**     |  **1.009$\pm$0.064**  |      **1.830$\pm$0.194**     |  **1.021$\pm$0.212**  |
> >
> >
> > - We can see that **DSPNET remains stable and continues to outperform baselines** under different unobserved confounding levels. The above results indicate that our method is robust even when there are unobserved confounders that can not be captured.

---

> ### Author Response · Authors · 2025-11-21
> **The response to Reviewer mUQv: [W2 And Q2]**
>
> > **W2 and Q2: Diagnostics of the Interference Representation.**
>
> **Response**: Thank you for your comments. We will clarify the role of the interference representation $\boldsymbol{e}_i^t$ from the following two points:
>
> **(1) Conceptual role of $\boldsymbol{e}_i^t$**
>
> - In many real-world settings, the interference experienced by a unit depends not only on the treatment assignments of its neighbors, but also on the **behavioral patterns** through which those neighbors engage with the intervention:
>   - For example, in a health intervention like jogging, a treated neighbor can influence a unit only if the neighbor actively engages in the activity and is inclined to share health-related behaviors.
>   - Thus, interference depends on **how treated neighbors behave**, not merely on how many neighbors are treated.
>
> - Consequently, unlike prior work that models exposure as a one-dimensional function of neighbors’ treatments (e.g., averaging or summing treatment values), our formulation uses a **learnable, treatment-gated representation** to capture richer forms of interference:
>
>  $$\boldsymbol{e_i^t}=\sum_{j \in \mathcal{ G}_i^t}d_j^t\cdot\boldsymbol{r}_j^t,
> $$
> where the neighboring  treatment assignment $d_j^t$, acts as gating signals, modulating the influence of neighbors’ latent behavioral representations $\boldsymbol{r}_j^t$.
>
>
> - This design allows DSPNET to model the **behavioral pathways** through which interference propagates, rather than collapsing spillover into a single scalar quantity. The resulting $\boldsymbol{e}_i^t$ can therefore be interpreted as a **data-driven embedding of the behavioral influence exerted by treated neighbors**.
>
>
> **(2) Additional comparison with scalar exposure summary**
>
> To further clarify the distinction between our behavior-based interference modeling and traditional scalar exposure mappings, we also compare our original modeling mechanism with the following two simplified scalar exposure summary functions:
>
> - **Sum-Pooling**: directly sum of the treatment values of unit’s neighbors, thus the exposure summary of unit $i$ is $\boldsymbol{e}_i^t=\sum\_{j \in \mathcal{G}_i^t}d_j^t$.
>
> - **Average-Pooling**: average the treatment values of unit’s neighbors, then the exposure summary of unit $i$ is $\boldsymbol{e}_i^t=\frac{\sum\_{j \in \mathcal{G}_i^t}d_j^t}{|\mathcal{G}_i^t|}$.
>
> - The empirical comparison results between the original modeling strategy with the two above variants on DSPENT are as follows:
>
> |     Variants    |          Flickr          |                  |        BlogCatalog       |                  |
> |:---------------:|:------------------------:|:----------------:|:------------------------:|:----------------:|
> |                 | $\sqrt{\epsilon_{PEHE}}$ | $\epsilon_{ATE}$ | $\sqrt{\epsilon_{PEHE}}$ | $\epsilon_{ATE}$ |
> |   Sum-Pooling   |     1.570 $\pm$ 0.190    | 1.163 $\pm$0.162 |     1.752 $\pm$0.166     |  1.183$\pm$0.182 |
> | Average-Pooling |     1.695 $\pm$ 0.152    | 1.142 $\pm$0.135 |     1.543 $\pm$0.164     | 1.073 $\pm$0.198 |
> |     Original    |     **1.497$\pm$ 0.145**     | **0.890$\pm$ 0.080** |      **1.464$\pm$0.119**     |  **0.845$\pm$0.105** |
>
> - We can see that both substitutions lead to drop in performance on CATE-ID estimation. This confirms that modeling only the count or fraction of treated neighbors is insufficient, and that the learned representation $\boldsymbol{e}_i^t$ indeed **captures meaningful behavioral pathways of interference** beyond simple treatment aggregation.

---

> ### Author Response · Authors · 2025-11-21
> **The response to Reviewer mUQv: [W3 and Q4]**
>
> > **W3 and Q4: Realism and Generalizability of the Evaluation. All experiments are synthetic with simulated treatments, outcomes, and interference. No semi-synthetic or real-world dataset. Are there any known limitations or scalability barriers when moving from synthetic to real data?**
>
> **Response**: Thank you for your comments. Below we clarify the realism of our evaluation setup and discuss the applicability and limitations of DSPNET on real-world dynamic networks.
>
> **(1) Realism of the (semi-synthetic) evaluation setup**
>
> We would like to clarify that our evaluation is **semi-synthetic**, not fully synthetic. Specifically:
>
> - **Real-world graph topology.** The network dynamic networks used in our evaluation are constructed from **real social-network graphs**, preserving realistic structural properties such as heavy-tailed degree distributions, community structure, and clustering patterns.
>
> - **Real-world covariates.** Node features are also **derived from real-world user metadata**, ensuring  that the covariates influencing treatment assignment and outcomes reflect actual heterogeneity observed in real social platforms.
>
> - **Established causal mechanism**. The generation of treatment, outcome, interference and temporal dependence are all generated following established principles:
>   - The temporal dependence follows a standard q-order autoregressive process (Mills, 1990; Melnychuk et al, 2022;  Bica et al, 2020).
>   - Treatment and outcome mechanisms follow the  widely used assumption and formulation in previous causal literature (Guo et al. 2020, Ma et al.2021, Huang et al. 2023, ).
>   - The mechanism through which interference is generated also aligns with real-world situations: it depends not only on neighbors’ treatment assignments but also on their behavioral patterns.
>
> - Thus, the evaluation combines **realistic network structure + realistic covariates + controlled causal mechanism**, enabling rigorous assessment of CATE-ID under plausible settings.
>
> **(2) Applicability and Limitation of DSPNET to real-world dynamic networks**
>
> - **Applicability**:
>
>   - Methodologically, DSPNET only requires a sequence of network snapshots, node-level covariates, time-varying treatments, and outcomes to train and inference.
>
>   - This setup is compatible with many real-world applications, such as recommendation or social-influence systems where personalized recommendations as treatments and engagement as outcomes.
>
>   - Thus, the proposed model DSPNET can, in principle, be applied without modification.
>
> - **Limitation**:
>   - Some assumptions adopted in the causal mechanism such as Ignorability can not be fully verified.
>
>   - Ground-truth CATE-ID is unavailable, this is why semi-synthetic evaluation, which is a standard in causal inference, remains necessary for testing those methods for treatment effect estimation.
>
>   - Behavioral interference mechanisms may be more complex, and real-world datasets may require richer covariates or domain-specific modeling choices.
>
>   - But we have to clarify that the above considerations reflect **general challenges** in network-based causal inference rather than limitations unique to DSPNET.

---

> ### Author Response · Authors · 2025-11-21
> **The response to Reviewer mUQv: [W4]**
>
> > **W4: Scalability untested.  There is no complexity or runtime comparison, and DSPNET’s feasibility on large-scale graphs is unclear.**
>
> **Response**: Thank you for your comments. Below we provide a detailed time-space complexity analysis of DSPENT and conduct an empirical scalability study:
>
> **(1) Time-space complexity analysis**
>
> - At each time step $t$, DSPNET mainly consists of the following four components:
>   - (i) GCN backbone with $L_g$ layers,
>   - (ii) an interference module that aggregates neighbor representations,
>   - (iii) a GRU-based temporal encoder,
>   - (iv) MLP backbone with $L_m$ layers for outcome and treatment prediction.
>
> - The time complexity of each component is as follows:
>   - **GCN backbone**. For a graph with $N=|V_t|$ nodes, $M_t$ edges, and hidden dimension $d_h$, a single sparse GCN layer first applies a linear transformation with complexity $\mathcal{O}(Nd_h^2)$, and then  performs sparse neighbor aggregation with complexity  $\mathcal{O}(M_td_h)$. Thus an $L_g$-layer GCN has complexity
> $\mathcal{O}(L_g(Nd_h^2+M_td_h)).$
>
>   - **Interference module**. The interference module aggregates neighbor representations with treatment signals. This is implemented as a sparse edge-wise aggregation and has complexity $\mathcal{O}(M_td_h)$.
>
>   - **GRU-based temporal encoder**. For each node, the GRU updates its hidden state using matrix multiplications of size $d_h \times d_h$. Over all nodes, this yields complexity $\mathcal{O}(Nd_h^2)$ per time step.
>
>   - **MLP backbone**. The potential-outcome and treatment-prediction heads are implemented as $L_m$-layer MLPs, each fully connected layer costs $\mathcal{O}(Nd_h^2)$, so the total complexity is $\mathcal{O}(L_mNd_h^2)$.
>
> - Combining these components over a  sequence of length $T$ assuming the edge size is of order $M_t=M$, the per-epoch training time complexity of DSPNET:
> $$
> \mathcal{O}(T((L_g+1)Md_h+(L_g+1+L_m)Nd_h^2))=\mathcal{O}(T(L_gMd_h+(L_g+L_m)Nd_h^2)),
> $$
> which is **linear in the number of edges $M$ and nodes $N$**.
>
>
> - **Space Complexity**: The space complexity is dominated by storing the sparse adjacency structure and node embeddings at each time step, i.e., $\mathcal{O}(M+Nd_h)$, which is **also linear in $M$ and $N$**.  In implementation, DSPNET processes the sequence in a streaming fashion and does not need to retain all time steps in memory simultaneously, which further improves practical scalability.
>
> **(2) Scalability investigation**
>
> - To provide quantitative evidence of scalability, we conducted an investigation evaluating how DSPNET’s runtime and memory usage change as the size of the dynamic network increases.
> - We constructed larger dynamic networks by scaling the number of nodes and edges to roughly **1×, 2×, 4×, 8× and 10×  of the original size** on Flickr and BlogCatalog. DSPNET was trained on each scaled dataset under the same hardware configuration, and we recorded the **wall-clock time** per epoch and peak **GPU memory usage**:
>
> |   Dataset   | Scale       | #Nodes | #Edges | Time per Epoch (s) | GPU Mem (GB) |
> |:-----------:|-------------|--------|--------|--------------------|--------------|
> |    Flickr   | $1 \times$  | 7.5K   | 239K   | 0.24s              | 2.8GB        |
> |             | $2 \times$  | 15K    | 478K   | 0.31s              | 5.6GB        |
> |             | $4 \times$  | 30K    | 956K   | 0.51s              | 11.3GB       |
> |             | $8 \times$  | 60K    | 1.91M  | 0.94s              | 22.5GB       |
> |             | $10 \times$ | 75K    | 2.39M  | 1.14s              | 28.1GB       |
> | BlogCatalog | $1 \times$  | 5K     | 171K   | 0.24s              | 1.7GB        |
> |             | $2 \times$  | 10K    | 342K   | 0.29s              | 3.4GB        |
> |             | $4 \times$  | 20K    | 684K   | 0.39s              | 6.8GB        |
> |             | $8 \times$  | 40K    | 1.36M  | 0.65s              | 13.2GB       |
> |             | $10 \times$ | 50K    | 1.71M  | 0.81s              | 16.5GB       |
>
>
> - Overall, the results show that both runtime and memory scale approximately linearly with graph size, **consistent with our theoretical complexity analysis**.
>
> - Importantly, DSPNET trains and infers stably even on graphs with **≈75K nodes and 2.39M edges**. For much larger graphs (e.g., industrial-scale networks), standard engineering optimizations—such as neighbor sampling or distributed GCN training—can be incorporated, as is common for GNN-based models in practice.

---

> ### Author Response · Authors · 2025-11-21
> **The response to  to Reviewer mUQv: [W5, W6 and Q3]**
>
> > **W5: No uncertainty estimation — The model reports point estimates without confidence intervals or statistical tests, which is critical in causal inference.**
>
> **Response**: Thank you for your comments. Following your suggestion, we conducted statistical tests to verify the significance of the improvements delivered by DSPNET.
> - We conduct the **paired t-test** to evaluate whether DSPNET significantly outperforms the strongest baseline, i.e., DNDC. The significance table under the setting of different dynamic levels is as follows:
>
> |             |               | $\sqrt{\epsilon_{PEHE}}$ |                  |           | $\epsilon_{ATE}$ |                  |           |
> |-------------|---------------|:------------------------:|:----------------:|:---------:|:----------------:|:----------------:|:---------:|
> | Dataset     | Dynamic Level |          DSPNET          |       DNDC       | $p$-value |      DSPNET      |       DNDC       | $p$-value |
> | Flickr      | 0.1%          |     1.497$\pm$ 0.145     |  2.589$\pm$0.959 |   <0.05   | 0.890$\pm$ 0.080 |  1.618$\pm$0.781 |   <0.05   |
> |             | 0.5%          |     2.062 $\pm$0.498     | 3.062$\pm$ 0.379 |   <0.05   | 1.144 $\pm$0.128 | 1.915$\pm$ 0.187 |   <0.05   |
> |             | 1.0%          |     2.189$\pm$ 0.205     | 3.291$\pm$ 0.522 |   <0.05   | 1.351$\pm$ 0.209 | 2.194$\pm$ 0.578 |   <0.05   |
> | BlogCatalog | 0.1%          |     1.464 $\pm$0.119     | 2.475 $\pm$0.462 |   <0.05   | 0.845 $\pm$0.105 | 1.454 $\pm$0.400 |   <0.05   |
> |             | 0.5%          |     1.506 $\pm$0.237     | 2.419 $\pm$0.332 |   <0.05   | 0.913 $\pm$0.204 | 1.319 $\pm$0.329 |   <0.05   |
> |             | 1.0%          |     2.227 $\pm$0.378     | 3.367 $\pm$0.757 |   <0.05   | 1.183 $\pm$0.290 | 1.723 $\pm$0.530 |   <0.05   |
>
> - The results indicate that the improvements of DSPNET over DNDC are **statistically significant** (p-values < 0.05, often much lower).
>
> > **W6 and Q3: Comparison with Prior Dynamic Network Models, Novelty over DNDC and SPENT. The novelty over DNDC (Ma et al., 2021) and SPNET (Huang et al., 2023) beyond architectural integration. A detailed ablation or theoretical justification isolating what enables DSPNET to outperform these baselines would help clarify its unique contribution.**
>
> **Response**: Thank you for your comments. The novelty of our method  goes beyond architectural integration and lies in the **causal problem formulation**, **identifiability analysis**, and the way the **architecture is tailored to handle dynamic interference**. We clarify this in four aspects:
>
> **(1) Different causal estimand and problem setting**.
>
> - DNDC focuses on dynamic deconfounding **without interference**, implicitly relying on SUTVA-like conditions. Its estimand is the individual treatment effect in a dynamic environment where one individual’s treatment does not affect the outcome of its neighbors.
>
> - SPNET models interference  but is designed for **static networks**, it can not address time-varying confounders and dynamic interference on the evolving graph structures.
>
> - Our model, DSPNET, is explicitly designed for dynamic networks with neighborhood interference. We introduce a **new estimand, CATE-ID**, where the outcome depends jointly on the individual treatment and a time-varying exposure $E_i^t$. This estimand and its associated causal mechanism are not covered by either DNDC or SPNET.
>
> **(2) New identifiability results under dynamic interference.**
>
> - Building on the new problem formulation and estimand, we formally provide **a new identifiability theorem (Theorem 3.3) for CATE-ID in the presence of dynamic interference** under some necessary assumptions.
>
> - However, neither DNDC nor SPNET studies identifiability of treatment effects in settings where both confounders and interference evolve over time.
>
> - Thus, the theoretical contribution of DSPNET is not simply reusing prior assumptions, but **extending identifiability to a new causal estimand and regime**.
>
> **(3) New architecture explicitly designed for modeling dynamic interference**
>
> - In real-world settings, interference is shaped not only by neighbors’ treatment assignments but also by **how those treatments interact with the neighbors’ intrinsic behavioral patterns**.
>
> - To address this, DSPNET uses treatment assignments as gating signals and constructs a learnable interference representation through an additional GCN, which acts as a flexible exposure summary function (see Eq. (6)). This design enables the model to **capture how treatments propagate through, and interact with latent behavioral patterns**.

---

> ### Author Response · Authors · 2025-11-21
> **Continue for [W6 and Q3]**
>
> **(4) Empirical ablation study**
>
> Actually we conducted an ablation study ( See Table 2) that isolates the contribution of each component by deriving different variants under default experimental setup,  and we can directly explain why DSPNET outperforms DNDC and SPNET based on the ablation results:
>
> | Variants |          Flickr          |                  |        Blogcatalog       |                  |
> |:--------:|:------------------------:|:----------------:|:------------------------:|:----------------:|
> |          | $\sqrt{\epsilon_{PEHE}}$ | $\epsilon_{ATE}$ | $\sqrt{\epsilon_{PEHE}}$ | $\epsilon_{ATE}$ |
> |  w/o IM  |        1.938±0.242       |    1.245±0.203   |        1.822±0.138       |    1.118±0.164   |
> |  w/o GRU |       10.235±1.768       |   6.854 ±1.014   |       10.652±0.718       |    3.547±0.912   |
> | Original |      **1.497±0.145**     |  **0.890±0.080** |      **1.464±0.119**     |  **0.845±0.105** |
>
>
> - **w/o IM**: This variant removes the interference modeling  module, reducing DSPNET toward a DNDC-like dynamic deconfounder. We can see that excluding  interference modeling leads to clear performance degradation, showing that explicitly modeling interference is crucial.
>
> - **w/o GRU**: This variant excludes the Gated Recurrent Unit, disabling the model’s ability to capture temporal dependencies and historical information, making the model close to SPNET. We can see that it leads to a substantial performance loss without capturing the temporal dependency.
>
>
>
>
> ---
>
> **References**:
>
> [1] Guo et al, Learning individual causal effects from networked observational data, WSDM 2020.
>
> [2] Huang et al, Modeling interference for individual treatment effect estimation from networked observational data, TKDD 2023.
>
> [3] Ma et al, Deconfounding with networked observational data in a dynamic environment, WSDM 2021.
>
> [4] Veitch et al, Using embeddings to correct for unobserved confounding in networks, NeurIPS 2019.
>
> [5] Hill et al,  Bayesian nonparametric modeling for causal inference, Journal of Computational and Graphical Statistics 2011.
>
> [6] Shalit et al, Estimating individual treatment effect: generalization bounds and algorithms, ICML 2017.
>
> [7] Yao et al, Representation learning for treatment effect estimation from observational data, NeurIPS 2018.
>
> [8] Melnychuk et al, Causal transformer for estimating counterfactual outcomes, ICML 2022.
>
> [9] Bica et al, Estimating counterfactual treatment outcomes over time through adversarially balanced representations, ICLR 2020.
>
> [10] Terence C Mills. Time series techniques for economists. Cambridge University Press, 1990.
>
> ---

---

> ### Author Response · Authors · 2025-11-26
> **Looking forward to your further feedback**
>
> Dear Reviewer mUQv,
>
> Once again, we are grateful for your time and effort for reviewing our manuscript and your positive evaluation of our work!
>
>  As our responses have been posted for a few days, we would like to kindly check whether they have sufficiently addressed your concerns. We would greatly appreciate any further feedback you may have, thank you so much!
>
> Best wishes,
>
> Authors

---

### Official Review · Reviewer_pbEV · 2025-10-30

**Soundness:** 2
**Presentation:** 2
**Contribution:** 1
**Rating:** 2
**Confidence:** 4

**Summary:**

The authors propose an end-to-end framework, DSPNET, which integrates GCN and RNN to represent dynamic confounders while simultaneously modeling neighborhood interference. The method is evaluated on semi-synthetic Flickr and BlogCatalog datasets.

**Strengths:**

The paper addresses an important problem of modeling causal effects under network interference in dynamic networks.

**Weaknesses:**

Weak novelty. The overall framework of this manuscript is highly similar to Ma, Jing et al., WSDM 2021 (DNDC: Deconfounding with Networked Observational Data in a Dynamic Environment), with only an additional “neighborhood interference” modeling component. However, the interference module itself is relatively simple, and the contribution is not substantial enough to meet the innovation standards expected at ICLR.

**Questions:**

1. The authors claim in the Reproducibility Statement that they have provided the source code to ensure reproducibility. However, the code link on the first page of the paper is invalid.

2. The authors assume the existence of a function Φ𝑧(⋅) that can capture all latent confounders (Assumption 3.1). This assumption may be too strong in realistic scenarios. It is suggested to include robustness or sensitivity analyses in the appendix to evaluate the impact of unobserved confounders that may not be fully captured.

3. The datasets used are static and semi-synthetic. It is unclear whether the equations used to generate treatment and outcome variables, as well as the way dynamics are manually introduced, are theoretically justified or empirically grounded. The authors should clarify how this data generation process may affect the validity of the results.

4. The paper lacks architectural transparency. The number of GCN and MLP layers, hidden dimensions, and epoch are not reported, and there is no analysis of how network depth or model complexity affects performance. Including these details would strengthen both reproducibility and clarity.

5. The scalability of the proposed model to large-scale dynamic networks (e.g., with millions of nodes) is unclear. The authors are encouraged to provide a quantitative time or space complexity analysis.

6. There is a typographical error: “Abalation” should be corrected to “Ablation.”

---

> ### Author Response · Authors · 2025-11-21
> **The response to Reviewer pbEV: [W1 and Q1]**
>
> We sincerely appreciate the reviewer’s great efforts and thoughtful comments, which greatly help improve our manuscript. Below, we address the concerns and questions point by point and try our best to update the manuscript accordingly, these updates are temporarily highlighted in $\textcolor{blue}{blue}$ to facilitate the reviewer’s checking.
>
> > **W1: Weak novelty. The overall framework of this manuscript is highly similar to Ma, Jing et al., WSDM 2021 (DNDC: Deconfounding with Networked Observational Data in a Dynamic Environment), with only an additional “neighborhood interference” modeling component.**
>
> **Response**: Thank you for your comments. We would like to clarify that our DSPNET is different from DNDC, in terms of the problem formulation, assumptions and the theoretical framework it enables. The novelty of this work does not lie in adding a small module to DNDC, but in introducing a **different causal problem, a new causal estimand, a new identifiability result, and an architecture specifically designed for dynamic interference**, which DNDC cannot handle by construction.
>
> **(1) Different Causal Problem**
>
> DSPNET solves a **different causal problem** that DNDC cannot address:
>   - **DNDC assumes no interference** that the treatment assignment of an individual would not causally affect the outcome of other individuals.
>   - **Our work extends to dynamic neighborhood interference**, where an individual’s treatment could affect the outcomes of its neighboring individuals dynamically.
>
> **(2) New Causal Estimand and Identifiability Result**
>
>
> - We **propose a new causal estimand CATE-ID by adopting new definition (i.e., Environment Exposure) and assumption (Assumption 2.2 for well-defined potential outcome under exposure)**, enabling the assessment of an intervention’s intrinsic causal effect on an individual while excluding dynamic interference.
>
> - In this case, CATE-ID’s identification and estimation can not be covered by DNDC. DNDC cannot model or identify effects under interference violating SUTVA assumption, while our work provides a **new identifiability theorem framework (Theorem 3.3)** tailored for interference and environment exposure in dynamic networks.
>
> **(3) New architecture specifically designed for modeling dynamic interference**
>
> The proposed interference modeling is not an add-on module but a core component which figures out the interference pattern in a learnable manner:
>
> - In real-world networks, interference depends not only on neighbors’ treatment assignments, **but also on how these treatments interact with intrinsic behavioral patterns of individuals**.
>
> - For example, a health intervention (e.g., jog) may affect a user only if its treated neighbors actively engage in and share health-related behaviors.
>
> - Hence, simple exposure summaries (e.g., averaging neighbor treatments: $\boldsymbol{e}_i^t=\frac{\sum\_{j \in \mathcal{G}_i^t}d_j^t}{|\mathcal{G}_i^t|}$) oversimplify these patterns.
>
> - Thus, unlike prior works, DSPNET **leverages treatment assignments as gating signals and builds a learnable interference representation** through an additional GCN module, which serves as the exposure summary function for modeling interference, as shown in Equation (6).
>
>
> > **Q1: The authors claim in the Reproducibility Statement that they have provided the source code to ensure reproducibility. However, the code link on the first page of the paper is invalid.**
>
> **Response**: Thank you for your comment.
>
> - We verified the anonymous code link on multiple machines (including tests by our colleagues), and **it has consistently worked on our side**. We suspect the issue may have been caused by temporary network instability or short-term maintenance of the anonymous website when you open the link.
>
> - To ensure smooth access, we have **updated the anonymous code link in the revised submission**.  Please kindly let us know if the reviewer still encounters any difficulty accessing the updated link.

---

> ### Author Response · Authors · 2025-11-21
> **The response to Reviewer pbEV: [Q2]**
>
> > **Q2: The authors assume the existence of a function Φ𝑧(⋅) that can capture all latent confounders (Assumption 3.1). This assumption may be too strong in realistic scenarios. It is suggested to include robustness or sensitivity analyses in the appendix to evaluate the impact of unobserved confounders that may not be fully captured.**
>
> **Response**: Thank you for your questions. We would like to clarify why we adopt  assumption 3.1 in our work and report the new results of robustness analyses of unobserved confounders.
>
> **(1) Why we adopt Assumption 3.1**
>
> - Assumption 3.1 states that although latent confounders cannot be directly observed, they can be inferred from observable covariates, network structure, and neighbor characteristics by a function $\Phi_z(\cdot)$, which is adopted by many previous causal literature in network settings (e.g.,Veitch et al, 2019;  Guo et al, 2020; Ma et al., 2021; Huang et al, 2023).
>
> - If **some latent confounders exist and there is no a function $\Phi_z(\cdot)$ that can capture or recover those latent confounders**, then the causal estimand of CATE is **not identifiable** in standard conditions. Thus, Assumption 3.1 is necessary for the identifiability of CATE-ID, analogous to the classical “no unmeasured confounding” assumption used in standard causal inference.
>
>
> - **In the case where there exist latent confounders which can not be captured, one can conduct the Rosenbaum’s sensitivity analysis (Rosenbaum et al, 2007) to evaluate the impact of these latent confounders that cannot be captured by $\Phi_z(\cdot)$.** Let $\pi(d_i^t|z_i^t)$ denote the nominal propensity score that a unit $i$ receives treatment $d_i^t$ under the inferred confounders ${z}\_i^t=\Phi_{z}(\boldsymbol{x}\_i^t, \boldsymbol{X}_{\mathcal{G}_i}^t, \mathcal{H}^{t})$, and let $\pi(d_i^t|z_i^t,u_i^t)$ denote the true propensity score with unmeasured latent confounder $u_i^t$. The amount of unmeasured confounding is controlled by a given sensitivity parameter $\Gamma \geq 1$, such that:
> $$
> \Gamma^{-1}
> \le
> \frac{(1-\pi(d_i^t \mid z_i^t))\pi(d_i^t \mid z_i^t,u_i^t)}
> {\pi(d_i^t \mid z_i^t)(1-\pi(d_i^t \mid z_i^t,u_i^t))}
> \le
> \Gamma,
> \quad \forall d_i^t,z_i^t,u_i^t.
> $$
> Given a value of $\Gamma$, one can derive the corresponding worst-case perturbation of propensities allowed by above bound and recompute the treatment effect contrast under that worst-case.  Then by gradually increasing $\Gamma$, we can  examine how sensitive our causal conclusions are to the unmeasured latent confounders. The Rosenbaum sensitivity analysis provides a principled way to quantify how violations of Assumption 3.1 might influence empirical findings.

---

> > ### Author Response · Authors · 2025-11-21
> > **Continue for [Q2]**
> >
> > **(2) Robustness analyses of unobserved confounders**
> >
> > To address the reviewer’s suggestion, we conducted robustness analyses regarding the unobserved confounders that may not be fully captured. Specifically, we explicitly select {10%, 20%, 30%} covariates that affects both treatment and outcome as the latent confounders, and these selected latent confounders are **not** included in the construction of  function  $\Phi_z(\cdot)$,  thereby simulating scenarios where the confounder modeling module fails to capture all confounders directly.
> >
> > - First, the CATE-ID estimation performance of  DSPNET for different levels of unobserved confounding is as follows:
> >
> > |                            |           Flickr           |                        |        BlogCatalog        |                      |
> > | :-------------------------: | :------------------------: | :---------------------: | :------------------------: | :-------------------: |
> > | **Level** | $\sqrt{\epsilon_{PEHE}}$ |   $\epsilon_{ATE}$   | $\sqrt{\epsilon_{PEHE}}$ |  $\epsilon_{ATE}$  |
> > |             10%             |      1.535 $\pm$0.133      |    0.911 $\pm$0.102    |      1.613 $\pm$0.188      |   0.949 $\pm$0.206   |
> > |             20%             |      1.694 $\pm$0.102      |    0.968 $\pm$0.081    |      1.697 $\pm$0.089      |   1.009 $\pm$0.064   |
> > |             30%             |      1.712 $\pm$0.129      |    1.112 $\pm$0.093    |      1.830 $\pm$0.194      |   1.021 $\pm$0.212   |
> > |        original (0%)        |  **1.497$\pm$ 0.145**  | **0.890$\pm$ 0.080** |   **1.464 $\pm$0.119**   | **0.845 $\pm$0.105** |
> >
> > - Besides, we also compare the CATE-ID estimation of DSPNET with those network-based baselines (NetEST, SPNET, Deconfounder, and DNDC) under different levels of unobserved confounding:
> >
> > |   Dataset   |     Model    |       Level = 10\%       |                   |       Level = 20\%       |                   |       Level = 30\%       |                   |
> > |:-----------:|:------------:|:------------------------:|:-----------------:|:------------------------:|:-----------------:|:------------------------:|:-----------------:|
> > |   Dataset   |     Model    | $\sqrt{\epsilon_{PEHE}}$ |  $\epsilon_{ATE}$ | $\sqrt{\epsilon_{PEHE}}$ |  $\epsilon_{ATE}$ | $\sqrt{\epsilon_{PEHE}}$ |  $\epsilon_{ATE}$ |
> > |    Flickr   | Deconfounder |     8.350 $\pm$ 1.324    | 4.807 $\pm$ 0.942 |     8.578 $\pm$ 1.375    | 4.950 $\pm$ 1.003 |     8.810 $\pm$ 0.998    | 5.216 $\pm$ 0.667 |
> > |             |    NetEST    |      6.982$\pm$1.121     |  1.484$\pm$0.204  |      7.172$\pm$1.259     |  1.590$\pm$0.189  |      7.538$\pm$1.375     |  1.610$\pm$0.219  |
> > |             |     SPNET    |      8.870$\pm$1.123     |  1.334$\pm$0.256  |      9.045$\pm$1.260     |  1.355$\pm$0.304  |      9.233$\pm$1.453     |  1.398$\pm$0.210  |
> > |             |     DNDC     |     2.637$\pm$0.8878     |  1.724$\pm$0.581  |      2.749$\pm$0.752     |  1.751$\pm$0.585  |      2.878$\pm$0.604     |  1.768$\pm$0.592  |
> > |             |    DSPNET    |      **1.535$\pm$0.133**     |  **0.911$\pm$0.102**  |      **1.694$\pm$0.102**     |  **0.968$\pm$0.081**  |      **1.712$\pm$0.129**     |  **1.112$\pm$0.093**  |
> > | BlogCatalog | Deconfounder |    13.231 $\pm$ 1.839    | 8.932 $\pm$ 1.212 |    13.773 $\pm$ 1.792    | 8.978 $\pm$ 1.280 |    13.906 $\pm$ 2.092    | 9.126 $\pm$ 1.483 |
> > |             |    NetEST    |      8.659$\pm$1.065     |  1.632$\pm$0.319  |      8.859$\pm$1.095     |  1.752$\pm$0.319  |      9.132$\pm$1.146     |  1.827$\pm$0.322  |
> > |             |     SPNET    |      9.725$\pm$1.546     |  2.227$\pm$0.838  |      9.882$\pm$1.346     |  2.681$\pm$0.846  |     10.379$\pm$1.475     |  2.782$\pm$0.943  |
> > |             |     DNDC     |      2.571$\pm$0.336     |  1.527$\pm$0.486  |      2.748$\pm$0.364     |  1.684$\pm$0.388  |      2.906$\pm$0.440     |  1.692$\pm$0.407  |
> > |             |    DSPNET    |      **1.613$\pm$0.188**     |  **0.949$\pm$0.206**  |      **1.697$\pm$0.089**     |  **1.009$\pm$0.064**  |      **1.830$\pm$0.194**     |  **1.021$\pm$0.212**  |
> >
> >
> >
> >
> > - We can see that **DSPNET remains stable and continues to outperform baselines** under different unobserved confounding levels. The above results indicate that our method is robust even when there are unobserved confounders.

---

> ### Author Response · Authors · 2025-11-21
> **The response to Reviewer pbEV: [Q3]**
>
> > **Q3: The datasets used are static and semi-synthetic. It is unclear whether the equations used to generate treatment and outcome variables, as well as the way dynamics are manually introduced, are theoretically justified or empirically grounded. The authors should clarify how this data generation process may affect the validity of the results.**
>
> **Response**: Thank you for raising this question. We address your questions from the following two aspects:
>
> **(1) Justification of data generation**
>
>  First we  would like to clarify that the data-generating process (See Appendix A) to generate dynamic confounders, treatments, interference and outcomes are **grounded and motivated in the established causal mechanism and the formulations from prior works**:
>
> - **Confounders**. The confounders $\boldsymbol{z}_i^t$ are generated through a combination of ($i$) historical latent states via a $q$–order autoregressive process (Mills, 1990; Melnychuk et al, 2022;  Bica et al, 2020), ($ii$) current neighbors’ topic-based features, and ($iii$) the unit’s own current covariates. This setting follows the dynamic confounding mechanism used in (Ma et al, 2021), and it encodes that the underlying state of a unit at time $t$ is shaped jointly by its **own history, its current attributes, and the current network environment**, which is exactly the type of dynamic confounding our method is intended to handle.
>
> - **Treatment**. Treatment $d_i^t$ is generated via a Bernoulli distribution whose probability is a softmax over the inner products between $\boldsymbol{z}_i^t$ and two centroids $\boldsymbol{r}_0^t$ and $ \boldsymbol{r}_1^t$ (Controlled and Treated) in the same topic space. This means  **units whose confounders are closer to the “treated centroid” are more likely to receive the treatment**. This setup follows the treatment mechanism adopted in many causal literature (e.g.,Veitch et al, 2019;  Guo et al, 2020; Ma et al., 2021; Huang et al, 2023; Guo et al, 2020) in network settings.
>
> - **Interference**. The interference $\mathcal{I}_i^t$ is constructed by taking the dot product between a randomly sampled vector and each neighbor’s topic embedding (i.e., ${\boldsymbol{w}_p^t}^Tf_L(\boldsymbol{x}_j)$), yielding a scalar that represents **how strongly that neighbor can influence unit $i$**. Then the influence is **activated only when the neighbor receives treatment**, ensuring that interference arises specifically from treated neighbors and depends on their characteristics, which aligns with our causal formulation about interference.
>
> - **Outcome**. The potential outcomes $y_i^t(1)$ and $y_i^t(0)$ are linear functions of ($i$) the confounder-based similarity scores $s_{i,0}^t$ and $s_{i,1}^t$ between $\boldsymbol{z}_i^t$ and two centroids $\boldsymbol{r}_0^t$ and $ \boldsymbol{r}_1^t$, capturing individual-level treatment response, ($ii$) the interference term  $\mathcal{I}_i^t$ plus Gaussian noise. This directly mirrors our theoretical setup, where the **outcome depends jointly on individual treatment and network exposure due to the interference**.

---

> ### Author Response · Authors · 2025-11-21
> **Continue for [Q3]**
>
> **(2) Empirical investigation using other data generation process**
>
> - To further address the reviewer’s question about how this data generation process may affect the validity of the results, we additionally considered a more complex, nonlinear potential-outcome model.  Concretely, instead of using a linear combination of  the similarity-based scores and interference, we generated the potential outcomes via nonlinear generation process as follows:
>
> $$
> y_i^t(1) = \kappa_1 \cdot tanh(s_{i,1}^t) +  \kappa_0 \cdot tanh(s_{i,0}^t)^2 +  \alpha (\mathcal{I}_i^t)^2 + \epsilon_y^t,
> $$
>
> $$
> y_i^t(0) = \kappa_0 \cdot tanh(s_{i,0}^t) + \kappa_0 \cdot(s_{i,0}^t)^2 + \alpha (\mathcal{I}_i^t)^2 +  \epsilon_y^t,
> $$
>
> where $ \kappa_0,  \kappa_1, \alpha$ are scaling factor, $ \epsilon_y^t$ is Gaussian noise.
>
> - Under the above nonlinear generation process, we repeat the simulation procedure 10 times and compare our DSPNET and the strongest dynamic-network baseline, i.e., DNDC, for CATE-ID estimation task under different levels of dynamics.  The comparison results are as follows:
>
> |   Dataset   |  Model |        p\% = 0.1\%       |                       |        p\% = 0.5\%       |                       |         p\% = 1\%        |                       |
> |:-----------:|:------:|:------------------------:|:---------------------:|:------------------------:|:---------------------:|:------------------------:|:---------------------:|
> |             |        | $\sqrt{\epsilon_{PEHE}}$ |    $\epsilon_{ATE}$   | $\sqrt{\epsilon_{PEHE}}$ |    $\epsilon_{ATE}$   | $\sqrt{\epsilon_{PEHE}}$ |    $\epsilon_{ATE}$   |
> |    Flickr   |  DNDC  |     4.149 $\pm$ 0.516    |   2.317 $\pm$ 0.547   |     5.012 $\pm$ 0.852    |   2.998 $\pm$ 0.773   |     5.262 $\pm$ 1.035    |   3.077 $\pm$ 0.574   |
> |             | DSPNET |   **3.397 $\pm$ 0.616**  | **1.373 $\pm$ 0.319** |   **3.774 $\pm$ 0.494**  | **1.780 $\pm$ 0.343** |   **3.977 $\pm$ 0.702**  | **1.766 $\pm$ 0.312** |
> | BlogCatalog |  DNDC  |     1.998 $\pm$ 0.193    |   1.137 $\pm$ 0.177   |     2.020 $\pm$ 0.214    |   1.204 $\pm$ 0.199   |     2.153 $\pm$0.223     |    1.163 $\pm$0.129   |
> |             | DSPNET |   **1.416 $\pm$ 0.217**  | **0.830 $\pm$ 0.096** |   **1.390 $\pm$ 0.191**  | **0.813 $\pm$ 0.164** |   **1.429 $\pm$0.233**   |  **0.836 $\pm$0.183** |
>
>
>
> -  We can see that DSPNET still **consistently outperforms DNDC** across different levels of dynamics. These results indicate that the effectiveness of DSPNET does **not** hinge on a particular specification of the data generation process.

---

> ### Author Response · Authors · 2025-11-21
> **The response to Reviewer pbEV: [Q4]**
>
> > **Q4: The paper lacks architectural transparency. The number of GCN and MLP layers, hidden dimensions, and epoch are not reported, and there is no analysis of how network depth or model complexity affects performance. Including these details would strengthen both reproducibility and clarity.**
>
> **Response**: Thank you for raising this important point. We fully agree that reporting architectural specifications and analyzing model complexity can strengthen both clarity and reproducibility.
>
> **(1) Architectural Transparency**
>
> - We have added all architectural details to the revised manuscript as shown in Appendix C.1, The following table summarizes the configuration of DSPNET:
>
> | Component                 | Layers | Hidden Dimensions | Activation  |                      Remarks                     |
> |---------------------------|:------:|:-----------------:|:-----------:|:------------------------------------------------:|
> | GCN Backbone              |    1   |         70        |    ReLU     |                         -                        |
> | MLP Backbone              |    2   |         70        |     ReLU    |                         -                        |
> | Recurrent Backbone        |    -   |         70        |      -      |               Gated Recurrent Unit               |
> | Treatment-MLP Backbone   |    2   |        100        |   Sigmoid   |       MLP Component of predicting treatment      |
> | Dimension of Confounders  |   --   |         70        |      --     |        Final representation of confounders       |
> | Dimension of Interference |   --   |         70        |      --     |         Final representation of exposure         |
> | Optimizer                 |   --   |         --        |      --     | Adam (learning rate = 4e-3, weight_decay = 1e-2) |
> | Other Hyperparameters     |   --   |         --        |      --     |   $\alpha=1$,$\beta=1$,$\omega=1e-4$  |
>
>
> **(2) Empirical Study of Model Complexity**
>
> - We conduct an empirical study to examine how increasing the architectural depth and width affects the performance of DSPNET. Specifically, we independently explored three key architectural hyperparameters about model complexity:
>   - the depth of the GCN backbone.
>   - the depth of the MLP backbone.
>   - the size of the hidden dimensions.
>
>
> - For each hyperparameter, we varied one factor while fixing the remaining two to their default settings. Concretely, we evaluated GCN depths of {1, 2, 3} layers, MLP depths of {3, 4, 5} layers, and hidden dimensions of {100, 150, 200}. The performance of DSPNET for CATE-ID estimation on Flickr and BlogCatalog in different architectural settings is as follows:
>
> |                 |       |          Flickr          |                  |        BlogCatalog       |                   |
> |:---------------:|:-----:|:------------------------:|:----------------:|:------------------------:|:-----------------:|
> |     Setting     | Value | $\sqrt{\epsilon_{PEHE}}$ | $\epsilon_{ATE}$ | $\sqrt{\epsilon_{PEHE}}$ |  $\epsilon_{ATE}$ |
> | # of GCN Layers |   1   |     1.497$\pm$ 0.145     | 0.890$\pm$ 0.080 |     1.464 $\pm$0.119     |  0.845 $\pm$0.105 |
> |                 |   2   |     5.299 $\pm$4.407     | 2.543 $\pm$2.811 |     6.293 $\pm$4.441     |  3.091 $\pm$3.803 |
> |                 |   3   |     15.703 $\pm$3.403    | 6.714 $\pm$2.826 |     16.868 $\pm$7.206    | 8.758 $\pm$ 7.103 |
> | # of MLP Layers |   3   |     1.455 $\pm$0.145     | 0.814 $\pm$0.110 |     1.355 $\pm$0.145     |  0.814 $\pm$0.110 |
> |                 |   4   |     1.571 $\pm$0.107     | 0.918 $\pm$0.116 |     1.371 $\pm$0.107     |  0.858 $\pm$0.116 |
> |                 |   5   |     1.416 $\pm$0.149     | 0.850 $\pm$0.116 |     1.486 $\pm$0.149     |  0.850 $\pm$0.116 |
> | # of Hidden Dim |  100  |     1.435 $\pm$0.121     | 0.887 $\pm$0.087 |     1.412 $\pm$0.110     |  0.812 $\pm$0.141 |
> |                 |  150  |     1.373 $\pm$0.103     | 0.836 $\pm$0.089 |     1.419 $\pm$0.221     |  0.772 $\pm$0.130 |
> |                 |  200  |     1.436 $\pm$0.159     | 0.844 $\pm$0.084 |     1.577 $\pm$0.433     |  0.827 $\pm$0.154 |
>
>   - The results show that the performance of DSPNET under different MLP layers and hidden dimensions remains stable across all tested configurations.
>   - However, adding more GCN layers leads to a clear performance drop due to the well-known **over-smoothing effect**, where deeper GCNs produce increasingly similar node representations and erase individual-level heterogeneity essential for accurate CATE estimation. Moreover, in many networks, 1-hop neighbors already provide sufficient information, while **deeper propagation introduces irrelevant signals from distant nodes**, further degrading performance.

---

> ### Author Response · Authors · 2025-11-21
> **The response to Reviewer pbEV: [Q5]**
>
> > **Q5: The scalability of the proposed model to large-scale dynamic networks is unclear. The authors are encouraged to provide a quantitative time or space complexity analysis.**
>
> **Response**: Thank you for your comment. Below we will provide a detailed time–space complexity analysis and  scalability investigation of DSPNET to address your concerns .
>
> **(1) Time-space complexity analysis**
>
> - At each time step $t$, DSPNET mainly consists of the following four components:
>   - (i) GCN backbone with $L_g$ layers,
>   - (ii) an interference module that aggregates neighbor representations,
>   - (iii) a GRU-based temporal encoder,
>   - (iv) MLP backbone with $L_m$ layers for outcome and treatment prediction.
>
> - The time complexity of each component is as follows:
>   - **GCN backbone**. For a graph with $N=|V_t|$ nodes, $M_t$ edges, and hidden dimension $d_h$, a single sparse GCN layer first applies a linear transformation with complexity $\mathcal{O}(Nd_h^2)$, and then  performs sparse neighbor aggregation with complexity  $\mathcal{O}(M_td_h)$. Thus an $L_g$-layer GCN has complexity
> $\mathcal{O}(L_g(Nd_h^2+M_td_h)).$
>
>   - **Interference module**. The interference module aggregates neighbor representations with treatment signals. This is implemented as a sparse edge-wise aggregation and has complexity $\mathcal{O}(M_td_h)$.
>
>   - **GRU-based temporal encoder**. For each node, the GRU updates its hidden state using matrix multiplications of size $d_h \times d_h$. Over all nodes, this yields complexity $\mathcal{O}(Nd_h^2)$ per time step.
>
>   - **MLP backbone**. The potential-outcome and treatment-prediction heads are implemented as $L_m$-layer MLPs, each fully connected layer costs $\mathcal{O}(Nd_h^2)$, so the total complexity is $\mathcal{O}(L_mNd_h^2)$.
>
> - Combining these components over a  sequence of length $T$ assuming the edge size is of order $M_t=M$, the per-epoch training time complexity of DSPNET:
> $$
> \mathcal{O}(T((L_g+1)Md_h+(L_g+1+L_m)Nd_h^2))=\mathcal{O}(T(L_gMd_h+(L_g+L_m)Nd_h^2)),
> $$
> which is **linear in the number of edges $M$ and nodes $N$**.
>
>
> - **Space Complexity**: The space complexity is dominated by storing the sparse adjacency structure and node embeddings at each time step, i.e., $\mathcal{O}(M+Nd_h)$, which is **also linear in $M$ and $N$**.  In implementation, DSPNET processes the sequence in a streaming fashion and does not need to retain all time steps in memory simultaneously, which further improves practical scalability.
>
>
> **(2) Scalability investigation**
>
> - To provide quantitative evidence of scalability, we conducted an investigation evaluating how DSPNET’s runtime and memory usage change as the size of the dynamic network increases.
> - We constructed larger dynamic networks by scaling the number of nodes and edges to roughly **1×, 2×, 4×, 8× and 10×  of the original size** on Flickr and BlogCatalog. DSPNET was trained on each scaled dataset under the same hardware configuration, and we recorded the **wall-clock time** per epoch and peak **GPU memory usage**:
>
> |   Dataset   | Scale       | #Nodes | #Edges | Time per Epoch (s) | GPU Mem (GB) |
> |:-----------:|-------------|--------|--------|--------------------|--------------|
> |    Flickr   | $1 \times$  | 7.5K   | 239K   | 0.24s              | 2.8GB        |
> |             | $2 \times$  | 15K    | 478K   | 0.31s              | 5.6GB        |
> |             | $4 \times$  | 30K    | 956K   | 0.51s              | 11.3GB       |
> |             | $8 \times$  | 60K    | 1.91M  | 0.94s              | 22.5GB       |
> |             | $10 \times$ | 75K    | 2.39M  | 1.14s              | 28.1GB       |
> | BlogCatalog | $1 \times$  | 5K     | 171K   | 0.24s              | 1.7GB        |
> |             | $2 \times$  | 10K    | 342K   | 0.29s              | 3.4GB        |
> |             | $4 \times$  | 20K    | 684K   | 0.39s              | 6.8GB        |
> |             | $8 \times$  | 40K    | 1.36M  | 0.65s              | 13.2GB       |
> |             | $10 \times$ | 50K    | 1.71M  | 0.81s              | 16.5GB       |
>
> - Overall, the results show that both runtime and memory scale approximately linearly with graph size, **consistent with our theoretical analysis**.
>
> - Moreover, DSPNET continues to train and infer reliably even on graphs with **75k nodes and 2.39 million edges**, suggesting that the framework is capable of handling larger dynamic networks in practice.

---

> ### Author Response · Authors · 2025-11-21
> **The response to Reviewer pbEV: [Q6]**
>
> > **Q6: There is a typographical error: “Abalation” should be corrected to “Ablation.”**
>
> **Response**: Thank you for pointing out the typographical error. We have corrected it in the revised version, and we will further review the manuscript to ensure correctness throughout.
>
> ---
>
> **We hope the above clarifications and discussions sufficiently address your concerns, and we would greatly appreciate it if you would like to kindly consider adjusting your score. We look forward to your insightful and constructive responses to further help us improve the quality of our work. Thank you!**
>
> ---
>
> ---
>
> **References**:
>
> [1] Guo et al, Learning individual causal effects from networked observational data, WSDM 2020.
>
> [2] Huang et al, Modeling interference for individual treatment effect estimation from networked observational data, TKDD 2023.
>
> [3] Ma et al, Deconfounding with networked observational data in a dynamic environment, WSDM 2021.
>
> [4] Veitch et al, Using embeddings to correct for unobserved confounding in networks, NeurIPS 2019.
>
> [5] Rosenbaum et al, Sensitivity analysis for m-estimates, tests, and confidence intervals in matched observational studies,  Biometrics 2007.
>
> [6] Melnychuk et al, Causal transformer for estimating counterfactual outcomes, ICML 2022.
>
> [7] Bica et al, Estimating counterfactual treatment outcomes over time through adversarially balanced representations, ICLR 2020.
>
> [8] Guo et al, Counterfactual evaluation of treatment assignment functions with networked observational data, SIAM 2020.
>
> [9] Terence C Mills. Time series techniques for economists. Cambridge University Press, 1990.
>
> ---

---

> ### Author Response · Authors · 2025-11-26
> **Looking forward to your further feedback**
>
> Dear Reviewer pbEV,
>
> Once again, we are grateful for your time and effort for reviewing our manuscript. As our responses have been posted for a few days, we would like to kindly check whether they have sufficiently addressed your concerns. To make our revisions easier to follow, we briefly summarize how we have addressed your main concerns:
>
> - For the novelty concern (W1), we clarified that our work addresses a **different causal problem**, introduces a **new causal estimand**, provides a **dedicated identifiability theorem** and an **architecture specifically designed for dynamic interference** that DNDC can not support.
>
> - For the justification of Assumption 3.1 (Q2), we (i) clarify that Assumption 3.1 is **commonly adopted** in networked causal inference, analogous to the standard  “no unmeasured confounding” assumption, and (ii) conduct **new robustness experiments** when Assumption 3.1 is violated where some latent confounders are not captured , showing that DSPNET remains stable and continues to outperform baselines under unobserved confounding.
>
> - For the justification of data generation (Q3), we (i) clarify that our DGP for dynamic confounders, treatments, interference, and outcomes are **directly motivated by and consistent with prior causal literature** on dynamic networks, (ii) add **new experiments** under a more complex, nonlinear outcome generation process, indicating that our model does not rely on a specific generation formula.
> - For architectural transparency (Q4), we (i) report **full architectural details**, and (ii) add a **systematic empirical study** varying GCN depth, MLP depth, and hidden dimension, with the detailed analysis for those results.
>
> - For the scalability and complexity analysis (Q5), we (i) provide a **theoretical formal time–space complexity analysis** showing that DSPNET scales linearly in the number of nodes and edges, and (ii) add a **scalability experiment**  by enlarging the dynamic networks up to 75K nodes and 2.39M edges, where both runtime and GPU memory grow approximately linearly and training remains practical.
>
> We would greatly appreciate any further feedback you may have, thank you so much!
>
> Best wishes,
>
> Authors

---

### Official Review · Reviewer_NsgB · 2025-11-01

**Soundness:** 3
**Presentation:** 4
**Contribution:** 3
**Rating:** 8
**Confidence:** 3

**Summary:**

The paper works on the observation study problem where all network interference, network dynamics, and hidden confounders are present in a dynamic network causal model. The work targets the conditional treatment effect CATE-ID and estimates the conditional probability densities of treatments, confounders, and potential outcomes through an RNN coupled with a GNN. This is a methodological paper with performance validated by thorough real data simulation.

**Strengths:**

- **Originality**: the work is novel in considering (i) dynamic network in causal inference, (ii) set of confounders that depends on the full history.
- **Quality**: the paper is well-written with clear explanation on the methodology and have solid empirical results.
- **Clarity**: the writing is very clear and easy to follow.
- **Significance** : the paper provides a general framework for causal effect estimation on dynamic networks. Although no contribution to the fundamental theory, the work provides a decent insights of deal with such complicated causal inference tasks

**Weaknesses:**

- The major estimand of the paper is to estimate the CATE-ID, which conditions on the realized treatments and exposures. It appears to be off-topic to most causal inference tasks as the usual goal is the **average** treatment effects under a given policy or a given stationary distribution.

**Questions:**

1. (related to the weakness). How can the method be generalized to other causal estimands of interest? For example, the policy value (average treatment effects under a certain stationary distribution of networks/treatments..)
2. Does the method require the network dynamics to be stationary?

---

> ### Author Response · Authors · 2025-11-21
> **The response to Reviewer NsgB: [W1 and Q1]**
>
> We sincerely appreciate the reviewer’s great efforts and thoughtful comments, which greatly help improve our manuscript. Below, we address the concerns and questions point by point and try our best to update the manuscript accordingly, these updates are temporarily highlighted in $\textcolor{blue}{blue}$ to facilitate the reviewer’s checking.
>
> > **W1 and Q1: The major estimand of the paper is to estimate the CATE-ID, which conditions on the realized treatments and exposures. It appears to be off-topic to most causal inference tasks as the usual goal is the average treatment effects under a given policy or a given stationary distribution. How can the method be generalized to other causal estimands of interest? For example, the policy value (average treatment effects under a certain stationary distribution of networks/treatments).**
>
> **Response**: Thank you for your comments. Below we will clarify why we focus on estimating CATE-ID, how our estimand relates to the conventional causal estimands such as ATE, and how our framework can be generalized.
>
> **(1) Why we focus on CATE-ID**
>
> Our targeted estimand, CATE-ID, is a **conditional individual-level effect** that depends on the realized covariates, treatments, and exposure in a dynamic network with interference. This choice is motivated by two reasons:
>
> - In many applied settings (e.g., recommendation, targeted intervention), the decision-maker is interested in **heterogeneous effects at the individual level**  in order to design personalized strategies, rather than only a single global ATE.
>
> - In the presence of interference and time-varying exposure, a well-defined conditional estimand such as CATE-ID provides a **local building block**: Once we can estimate individual-level effects, we can aggregate them under various distributions or policies for other estimand of interest (e.g., ATE) .
>
> **(2)  From CATE-ID to other estimand**
>
> - For example, given an estimator of CATE-ID $\hat{\tau}_i^t$, the average treatment effect (ATE) under a distribution $\mathcal{D}$ can be obtained by averaging: $
> ATE_D=\mathbb{E}_D[\hat{\tau}_i^t]$.
>
> - Actually, we also report the metric *Mean Absolute Error on the Average Treatment Effect* ($\epsilon_{ATE}$), which measures how accurately the CATE-ID estimator recovers the **population-level ATE** under the overall data-generating distribution.
>
> - We further evaluate *Rank-weighted Average Treatment Effect* (RATE) to assess whether the CATE-ID estimates lead to **effective treatment prioritization rules** that benefits the population which is another way to measure the policy value derived from estimated CATE-ID.
>
> **(3)  Potential extension to more general estimands**
>
> Extending our framework to other estimands, such as the value of a specific treatment policy under a stationary distribution of networks and treatments, is conceptually straightforward:
>
> - A policy induces a distribution over $(\boldsymbol{X}^t, \boldsymbol{D}^t, \boldsymbol{Y}^t, \mathcal{H}^t, \boldsymbol{A}^t) \sim \tilde{\mathcal{D}}$;
> - Given an estimator of CATE-ID, one can compute or approximate the corresponding policy value by integrating the estimated CATE over that induced distribution $\tilde{\mathcal{D}}$.

---

> ### Author Response · Authors · 2025-11-21
> **The response to Reviewer NsgB: [Q2]**
>
> > **Q2: Does the method require the network dynamics to be stationary?**
>
> **Response**: Thank you for this thoughtful question. We would like to clarify that our method **does not require the network dynamics to be stationary** in the strict sense from the following aspects:
>
> **(1) No stationarity assumption in our causal formulation**
> - Our causal problem and identifiability analysis are formulated at each time step $t$, thus they are **irrelevant to whether the distributions across different time periods are stationary**.
> - Furthermore, the assumptions we adopted in this work, such as the Extended Ignorability and Consistency assumptions, are also **stated for current time step** $t$, and do not impose that the joint distribution  $(\boldsymbol{X}^t, \boldsymbol{D}^t, \boldsymbol{Y}^t, \mathcal{H}^t, \boldsymbol{A}^t)$ is stationary across time.
>
> **(2) Our model architecture is designed to handle non-stationary dynamics**
> - DSPNET uses a GCN backbone to encode graph snapshot of current time step and a GRU-based temporal module to aggregate historical information. This recurrent component is precisely introduced to capture **time-varying patterns** in latent confounders, rather than relying on stationarity.
>
> - While parameters are shared across time steps, this **does not imply or enforce a stationary process**. It simply assumes that a **common parametric family can approximate the evolving dynamics**.
>
> **(3) Experimental setup**
> - In our empirical evaluation, the dynamic networks are not strictly stationary. The adjacency matrix evolves over time via different degrees of random edge additions and deletions, and the  latent confounders follow q-order autoregressive process  (Mills, 1990; Melnychuk et al, 2022;  Bica et al, 2020) with noise, which may induce a moderate level of drift.
> - From the empirical results, we can see that the proposed DSPNET maintains strong performance under such evolving dynamics across different experimental settings (e.g., varying dynamic levels and interference strengths).
>
> ---
>
> **References:**
>
> [1] Terence C Mills. Time series techniques for economists. Cambridge University Press, 1990.
>
> [2] Melnychuk et al, Causal transformer for estimating counterfactual outcomes, ICML 2022.
>
> [3] Bica et al, Estimating counterfactual treatment outcomes over time through adversarially balanced representations, ICLR 2020.
>
>
> ---

---

> ### Author Response · Authors · 2025-11-26
> **Looking forward to your further feedback**
>
> Dear Reviewer NsgB,
>
> Once again, we are grateful for your time and effort for reviewing our manuscript and your positive evaluation of our work!
>
>  As our responses have been posted for a few days, we would like to kindly check whether they have sufficiently addressed your concerns. We would greatly appreciate any further feedback you may have, thank you so much!
>
> Best wishes,
>
> Authors

---

### Official Review · Reviewer_kQia · 2025-11-05

**Soundness:** 3
**Presentation:** 2
**Contribution:** 3
**Rating:** 8
**Confidence:** 3

**Summary:**

This paper focuses on the treatment effect estimation in dynamic network environments where both time-varying hidden confounders and network interference exist. The authors propose a new treatment effect estimand designed for dynamic settings with interference and introduce a new causal inference framework that explicitly models both time-varying hidden confounders and network interference based on graph nerual networks. To mitigate confounding bias in observational data, the proposed framework uses adversarial learning to enforce balance in the learned confounder representations. The framework is designed to accurately estimate causal effects in settings where the network structure and confounding evolve over time.

**Strengths:**

1. The paper highlights an important and underexplored perspective, estimating causal effects in dynamic network environments, which has received relatively little attention despite its relevance.
2. The proposed framework, based on graph neural networks, is well defined and effectively demonstrates the advantages of learning representations of hidden confounders over time and estimating treatment effects in dynamic networks.
3. This paper is very well-organized, easy to follow, and clearly presents its contributions.

**Weaknesses:**

1. The paper lacks clarity on how the proposed method addresses the entanglement between network structures and covariates. A more detailed explanation—whether through theoretical justification or intuitive examples—would help clarify how the model disentangles these dependencies and ensures valid causal estimation.

**Questions:**

1. Can the proposed approach be extended to settings where the underlying network structure is partially or entirely unobserved? Exploring the applicability of the proposed framework in such scenarios may offer a valuable direction for future research.

---

> ### Author Response · Authors · 2025-11-21
> **The response to Reviewer kQia: [W1]**
>
> We sincerely appreciate the reviewer’s great efforts and thoughtful comments, which greatly help improve our manuscript. Below, we address the concerns and questions point by point and try our best to update the manuscript accordingly, these updates are temporarily highlighted in $\textcolor{blue}{blue}$ to facilitate the reviewer’s checking.
>
> > **W1: The paper lacks clarity on how the proposed method addresses the entanglement between network structures and covariates. A more detailed explanation—whether through theoretical justification or intuitive examples—would help clarify how the model disentangles these dependencies and ensures valid causal estimation**.
>
> **Response**: We thank the reviewer for raising this subtle and important point. We note that the interaction between network structure and covariates is a well-known challenge in observational network analysis.  In particular, the causal literature has long recognized that similarity in covariates, link formation, and peer influence can be highly intertwined. We summarize below how existing literature frames this issue and how our work positions itself relative to it.
>
> **(1) Homophily due to covariates and  Contagion**
> - In observational social networks, node covariates and network structure are indeed strongly entangled, most prominently through **homophily** ( individuals with similar covariates are more likely to connect).
>
> - As shown by Shalizi et al (2011), homophily and contagion (peer influence/interference) are generically confounded, making it fundamentally impossible to fully distinguish **similarity-from-homophily vs. similarity-from-contagion** without experimental interventions or very strong structural assumptions under observed networks..
>
> - Prior work such as McFowland et al (2018) attempts to separate these two effects via latent space models, but this requires restrictive assumptions on the network generative process.
>
>
> **(2) Our modeling stance and what DSPNET does attempt**
>
> - In light of the above results, the goal of DSPENT is **not to disentangle homophily from contagion**, which is beyond the scope of this work.
>
> -  Instead, we explicitly assume a contagion-like interference mechanism, focusing on estimating treatment effects under peer influence given the observed dynamic networks. Specifically, network structure and covariates are treated as observed contextual information, and DSPNET learns:
>   - a representation of $Z_i^t$ to model the dynamic latent confounders by temporal  network structure and covariates.
>   -  an interference representation $\boldsymbol{e}_i^t$ to model how treated neighbors exert interference.
>
> - In other words,our work follows the contagion-style mechanism for interference and does not claim to identify or separate homophily from contagion, which would **require much stronger assumptions** than those made in this work.
> - Our identifiability analysis characterizes when CATE-ID can be recovered from the observed dynamic network with interference, and DSPNET is designed to approximate the required representations for valid causal estimation in this setting.
>
> Overall, we acknowledge that fully separating homophily-driven similarity from contagion-driven influence in dynamic observational networks is an open and challenging problem, and we have updated the above discussion in the limitation section (See Appendix D). Thank you so much for this insightful comment.

---

> > ### Author Response · Authors · 2025-11-21
> > **The response to Reviewer kQia: [Q1]**
> >
> > > **Q1: Can the proposed approach be extended to settings where the underlying network structure is partially or entirely unobserved? Exploring the applicability of the proposed framework in such scenarios may offer a valuable direction for future research.**
> >
> > **Response**: Thank you for this insightful question. We agree that scenarios with partially observed or entirely unobserved network structure are of substantial practical interest. Conceptually, our framework can be extended to such settings by treating the network structure as an additional latent object to be inferred rather than as a fixed input. Concretely:
> >
> > - In the **partially observed** case, one could combine the observed edges with a learned latent adjacency that is inferred from node features and temporal behavior, using graph structure learning or neural relational inference techniques. The resulting graph (observed + learned) could then be fed into DSPNET as the dynamic backbone for confounder and interference modeling.
> >
> > - In the **fully unobserved** case, one could first learn a latent dynamic graph from sequences of covariates, treatments, or outcomes (e.g., via a separate structure-learning module), and then apply DSPNET on top of this inferred network. In this view, DSPNET provides the causal estimation layer, while structure-learning methods provide an estimate of the underlying network topology.
> >
> > Overall, incorporating network structure uncertainty and jointly learning structure and interference-aware representations for CATE-ID estimation is a promising but nontrivial extension. We appreciate the reviewer for pointing it out explicitly.
> >
> >
> > ---
> >
> > **References**:
> >
> > [1] Shalizi et al, Homophily and contagion are generically confounded in observational social network studies." Sociological methods & research.
> >
> > [2] McFowland  et al,  Estimating causal peer influence in homophilous social networks by inferring latent locations." Journal of the American Statistical Association.
> >
> > ---

---

> ### Author Response · Authors · 2025-11-26
> **Looking forward to your further feedback**
>
> Dear Reviewer kQia,
>
> Once again, we are grateful for your time and effort for reviewing our manuscript and your positive evaluation of our work!
>
>  As our responses have been posted for a few days, we would like to kindly check whether they have sufficiently addressed your concerns. We would greatly appreciate any further feedback you may have, thank you so much!
>
> Best wishes,
>
> Authors

---

### Meta-Review · Area_Chair_Rgww · 2026-01-07

**Summary:**

Limited novelty: The proposed framework is very similar to DNDC (WSDM’21), with the neighborhood interference component viewed as relatively simple. Additional concerns were raised regarding reproducibility, justification of the data generation process, architectural transparency, and scalability.

Strength of identifiability assumptions: The realism of the extended ignorability and consistency assumptions was questioned, in particular the assumption that the learned representation can fully capture latent confounders along with the lack of empirical verification or sensitivity analysis in the original submission.

Evaluation realism and generalizability: The evaluation relies on semi-synthetic settings, which provides limited evidence for generalization to real-world dynamic networks, and the data generation process required clearer justification.

**Reviewer Concerns:**

Limited novelty: The rebuttal clarified differences from DNDC by emphasizing the dynamic interference setting, the new estimand (CATE-ID), and a dedicated identifiability result, and addressed concrete issues by adding architectural details, ablations, scalability analysis, and reproducibility fixes. However, whether these differences amount to sufficient novelty remains a matter of judgment.

Strength of identifiability assumption: The rebuttal justified the extended ignorability and consistency assumptions by linking them to prior work and added robustness experiments under unobserved confounding, which addressed the lack of empirical verification noted in the reviews. The assumptions themselves remain strong and not directly testable in practice.

Evaluation realism and generalizability: The rebuttal clarified the semi-synthetic evaluation setting, provided stronger justification of the data generation process, and added nonlinear outcome experiments. Nonetheless, evaluation on fully real-world outcomes is still missing, although this limitation is acknowledged as inherent to causal effect estimation.

**Reviewer Scores:**

Overall, the rebuttal directly addressed most technical and clarification-related concerns.
Reviewers who were initially positive would likely maintain their scores or gain increased confidence, while a borderline reviewer would likely shift toward a more favorable evaluation given that the main concerns were substantially addressed.
In contrast, although many concrete issues raised by the most critical review were improved, the fundamental concern about novelty is judgment-based and may not be fully resolved, which makes a major score reversal less likely.

---

### Decision · Program_Chairs · 2026-01-26

Accept (Poster)